# EVA-MVC: Equitable View-weight Allocation for Generic Multi-View Clustering

## ABSTRACT

Contemporary datasets sourced from the web often adopt a multi-view format, collecting data from diverse sources, domains, or modules. Existing methodologies employed to analyze such datasets frequently overlook or inaccurately allocate the view-weights, pivotal metrics reflecting each view's significance. This work introduces EVA-MVC, a simple yet effective algorithm designed for Equitable View-weight Allocation (EVA) seamlessly integrated with arbitrary Multi-view Clustering (MVC) methods. Within the EVA module, we establish theoretical connections between view supplementary and Multi-view Subspace Learning (MSL), leading to the partition of views into View Communities (VCs) based on these foundational principles. These VCs exhibit internal supplementarity similarities, facilitating Equitable View-weights Allocation through VC-specific MSL. The proposed EVA process precedes and operates independently of traditional or SOTA MVC approaches, requiring no additional processing or specialized design, making it an ideal preprocessing step for MVC applications. Through comprehensive evaluations across diverse multi-view datasets, our findings reveal that our EVA significantly enhances the effectiveness of mainstream MVC frameworks, resulting in a notable performance improvement.

## CCS CONCEPTS

• **Computing methodologies → Cluster analysis**.

## KEYWORDS

Multi-view Clustering, Multi-view Subspace Learning, Equitable View-weight Allocation, Consistency, Supplimentarity

**ACM Reference Format:**

Anonymous Author(s). 2024. EVA-MVC: Equitable View-weight Allocation for Generic Multi-View Clustering. In *Proceedings of ACM Conference (Conference'17)*. ACM, New York, NY, USA, 15 pages. https://doi.org/10.1145/nnnnnnn.nnnnnnn

## 1 INTRODUCTION

The landscape of data collection and acquisition methods has experienced significant diversification, leading to the development of various feature extraction techniques tailored to distinct data sources [5, 9, 46]. As each technique is typically designed with a specific data perspective, achieving a comprehensive representation of the data necessitates the integration of multiple views.

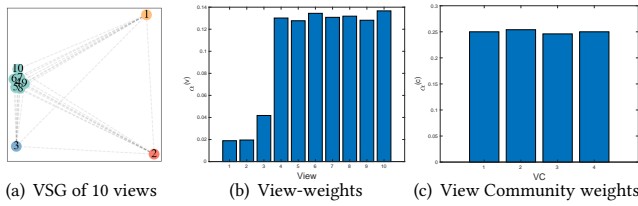

(a) VSG of 10 views  (b) View-weights  (c) View Community weights

**Figure 1: Illustration of the "Redundant Supplementarity" Issue on a Syntactic Multi-view Dataset: (a) View Supplementarity Graph (VSG), (b) View-weights Allocation via Multi-view Subspace Learning (MSL), and (c) View Community Weights Allocation via MSL. The issue is highlighted by a significant overlap among the final 7 views in the VSG. In (b), this issue is apparent as these views are assigned notably higher values, resulting in their dominance in the MSL. In (c), our method divides the VSG into 4 View Communities, achieving a more Equitable View-weight Allocation (EVA).**

For example, online news articles may contain multiple views, such as video, text, and images [32]. In the case of images, techniques like local binary patterns (LBP), histograms of oriented gradients (HOG), and Gabor descriptors can extract additional features for analysis and representation. Thus, effectively fusing diverse information is paramount in multi-view scenarios.

Multi-view Clustering (MVC) plays a crucial role in revealing the intrinsic structure across multiple views, finding widespread acceptance in data mining applications [13, 16, 26]. The effectiveness of MVC is guided by two fundamental principles: **Consistency** and **Supplementarity** [7, 34, 52]. Consistency aims to maximize alignment between different views, while supplementarity seeks for argumenting the unique information carried by each view that is not captured by others. Nonetheless, current research often exhibits a tendency to prioritize consistency over supplementarity.

In Multi-view Subspace Learning (MSL) [17, 29, 34], a prevalent MVC solution, the emphasis typically lies solely on the consistency principle. To achieve robust clustering [24, 45, 58], a lower-dimensional, more consistent subspace is learned by mapping the high-dimensional multi-view data into it [49], followed by conducting subsequent clustering operations within this refined subspace.

Unfortunately, Supplementarity, often lacking a robust theoretical foundation, tends to be disregarded by existing MVC methodologies. In this study, we introduce a formal definition of supplementarity within the MSL framework. This is done to show that, like Consistency, Supplementarity is quantifiable and comparable. Nevertheless, views with similar levels of supplementarity frequently dominate the MSL process, resulting in what we refer to as "Redundant Supplementarity". This phenomenon results in an over-reliance on specific views, significantly diminishing the representational capability of the learned subspace.

To illustrate "Redundant Supplementarity", we craft a synthetic dataset featuring 10 views derived from YTF100 (initially comprising 4 views) by replicating the final view six times. As illustrated in Figure 1 (a), these 10 views are depicted as nodes within a View Supplementarity Graph (VSG), where edges denote supplementarity similarities. Notably, the last seven views exhibit a substantial degree of supplementarity likeness. Utilizing a conventional MSL technique, the allocated view-weights ($\alpha^{(v)}$) for the 10 views are shown in Figure 1 (b). The weights attributed to the final 7 views significantly outweigh those assigned to the initial 3 distinct views. This misallocation of weights disregards the contributions of the initial 3 views. These observations highlight the importance and urgency of developing an Equitable View-Weight Allocation (EVA) mechanism in the domain of Multi-view Clustering (MVC).

In summary, this work contributes in the following folds:

(1) **Theoretical Foundation**: We propose a novel method, Equitable View-weight Allocation (EVA), theoretically grounded in the concept of Supplementarity. EVA effectively resolves the issue of "Redundant Supplementarity", thereby enhancing subsequent tasks such as clustering.

(2) **General**. Our EVA is compatible with arbitrary Multi-view Clustering (MVC) methods. Experiments show that EVA enhances the performance of both traditional or cutting-edge MVC methods, with a remarkable improvement.

(3) **Scale and Efficient**. Our method operates on minimal anchor graph, foregoing the need for full pairwise graph. Theoretical analyses affirm that applying EVA to the anchor graph yields results equivalent to those achieved with the full graph, ensuring overall efficiency and scalability.

(4) **Fast and Effective**. Through experiments on real and synthetic datasets, we demonstrate the performance of our framework in tackling the challenges of MVC.

## 2 PRELIMINARIES & RELATED WORK

Consider a multi-view dataset describing $n$ data points in $V$ views, denoted as $\mathcal{V} = \{\mathcal{V}^{(1)}, \cdots, \mathcal{V}^{(V)}\}$, where the view specific feature matrices are denoted as $X = \{X^{(1)}, \cdots, X^{(V)}\}$. For the feature matrix of the $v$-th view, $X^{(v)} \in \mathbb{R}^{m^{(v)} \times n}$, its $i$-th row and the $j$-th column of $X^{(v)}$ are denoted by $x_{i:}^{(v)}$ and $x_{:j}^{(v)}$, respectively. $Tr(X^{(v)})$ denotes the trace of $X^{(v)}$, $(X^{(v)})^T$ denotes its transpose, and its Frobenius norm is denoted by $\left\| X^{(v)} \right\|_F$. $\mathbf{1}$ and $I$ denote a column vector of ones and the identity matrix, respectively.

### 2.1 Multi-view Subspace Learning (MSL)

MSL assumes that high-dimensional multi-view data can be represented as a combination of multiple low-dimensional subspaces. These subspaces are discovered by using the original multi-view data as a reference or dictionary, with the objective of preserving the inherent structure found in the self-representation matrices. Mathematically, the overall framework of MSL aims to minimize the reconstruction loss and can be expressed as follows:

$$\min_{U,Y} \sum_{v=1}^{V} \alpha^{(v)} \|X^{(v)} - U^{(v)}Y\|_F^2 + \omega(Y),$$ (1)

$$s.t. \ \|U_{:,j}\|_2^2 \leq 1,$$

where $U^{(v)}$ and $Y$ represent the mapping models and the consistency (consistent representation). $\alpha^{(v)}$ denotes the $v$-th view-weight allocated by MSL. The term $\omega$ represents the consensus regularization term, which helps in training a global graph across the views. In the framework described, several MSL methods [31, 37, 48, 56] have been proposed to partition multi-view datasets by capturing their global structure.

### 2.2 Non-MSL Methods in MVC

In addition to MSL methods, there are other approaches in MVC that fall under the category of non-MSL methods. We will briefly introduce their ideas and representative methods.

**Multi-view Kernel Learning (MKL)**. MKL utilizes multiple kernel learning techniques [18, 19] for clustering. Liu et al. [27] used contrastive learning for KML, while ignore the supplementarity across multiple views.

**Multi-view Matrix Factorization Learing (MMFL)**. MMFL aims to enhance the learned representation by Non-negative Matrix Factorization (NMF) technology. Zong et al. [61] proposed a multi-manifold regularized NMF framework for preserving locally geometrical structure. Li et al. [25] designed a unified NMF framework to improve the representation quality. Zheng et al. [60] integrated NMF and $k$-means as a unified framework. In MV-Co-VH [11], NMF is performed on both visible and hidden views. Yang et al. [53] proposed to fuse multi-view data into a low-dimensional consensus embedding by NMF directly for efficiency.

**Multi-view Graph Learning (MGL)**. MGL utilizes graphs for describing multi-view dataset. Zhang et al. [59] introduced a MGL framework to combine several tasks including MVC. Wang et al. [41] proposed to generate a graph for each view and then fuse these graphs. Liu et al. [28] designed a plug-and-play anchor enhancement strategy to assist the MVC.

The methods discussed offer diverse strategies for fusing multi-view data, either treating views equally or assigning view-weights (potentially facing Redundant Supplementarity, as discussed in Section 3). Table 1 summarizes mainstream MVC methods, detailing their time and space complexities. Furthermore, our EVA module enhances these methods, marking "NA" where the source code is unavailable, on five real-world datasets affected by Redundant Supplementarity. The average improvement in Accuracy (ACC) is reported, with detailed experimental results accessible in the Evaluation and Appendix Sections.

## 3 PROPOSED FRAMEWORK: EVA-MVC

This section presents the EVA-MVC framework, outlining its motivation, methodology, optimization, and complexity analysis. For a comprehensive understanding of EVA-MVC, please refer to Figure 2, which comprises two modules: Equivalent View-weight Allocation (EVA) and Multi-view Clustering (MVC). EVA, the core of the framework, aims to tackle the issue of Redundant Supplementarity.

### 3.1 Redundant Supplementarity

MSL methodologies are founded on an process integrating multiple views into a unified low-dimensional subspace, denoted as $S$ [38] and expressed mathematically as:

$$S = \{X^{(v)} \in X : X^{(v)} = U^{(v)}Y + \mu^{(v)}\}$$ (2)

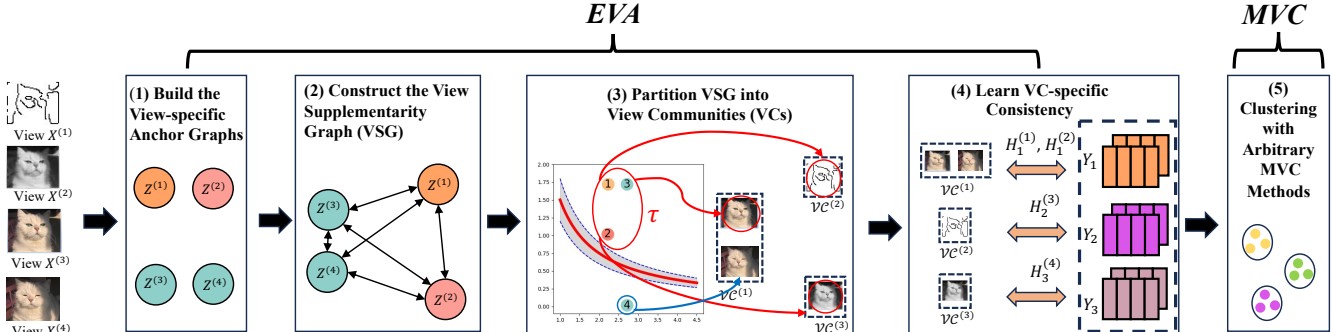

**Figure 2: Overview of the EVA-MVC Framework, comprising two modules: (1) Equitable View-weight Allocation (EVA) and (2) Multi-view Clustering (MVC). In EVA, views are partitioned into distinct View Communities (VCs) based on their supplementarity similarities, denoted as $\mathcal{VC}$. Subsequently, for each VC $\mathcal{VC}^{(c)} \in \mathcal{VC}$, a VC-specific consistency (consistent representation) $Y_c$ is learned. In MVC, these consistencies serve as inputs for an arbitrary MVC method.**

**Table 1: Overview of Mainstream MVC Methods.**

| MVC Method (Year) | MVC Class | Time & Space Complexity | EVA-Driven Enhancement (Avg. ACC on Five Datasets) |
|---|---|---|---|
| BMTMVC [59](2016) | MGL | $O(n^2) : O(n^2)$ | N/A |
| MCLES [6](2020) | MSL | $O(n^2) : O(n^2)$ | 5.6% |
| GMC [41](2022) | MGL | $O(n^2) : O(n^2)$ | N/A |
| FPMVS [44](2021) | MSL | $O(n) : O(n)$ | 5.8% |
| MSGL [23](2021) | MSL | $O(n) : O(n^2)$ | 3.5% |
| SMVSC [35](2021) | MSL | $O(n) : O(n)$ | 10.1% |
| SDMVC[51](2021) | MSL | $O(n) : O(n)$ | N/A |
| OMSC [8](2022) | MSL | $O(n) : O(n)$ | 15.3% |
| MV-Co-VH [11](2022) | MMFL | $O(n^2) : O(n^2)$ | N/A |
| CTMSC [10](2022) | MSL | $O(n^3) : O(n^2)$ | N/A |
| CCNMF [25](2023) | MMFL | $O(n^2) : O(n^2)$ | N/A |
| SMGC [36](2023) | MGL | $O(n^2) : O(n^2)$ | N/A |
| K-MCILSS [54](2023) | MKL | $O(n^3) : O(n^2)$ | N/A |
| LMGEC [14](2023) | MGL | $O(n) : O(n)$ | N/A |
| MERLIN [55](2023) | MSL | $O(n) : O(n)$ | N/A |
| MSC²D[21](2023) | MSL | $O(n^2) : O(n^2)$ | N/A |
| MFK [60](2023) | MMFL | $O(n^3) : O(n^2)$ | N/A |
| AWMVC [39](2023) | MMFL | $O(n) : O(n)$ | 9.76% |
| FastMICE [20](2023) | MGL | $O(n) : O(n)$ | 5.36% |
| CMVC [57](2024) | MGL | $O(n) : O(n)$ | 3.8% |

Here, $\mu^{(v)}$ represents the overall loss incurred while mapping the feature matrix $X^{(v)}$ of the $v$-th view to the subspace $S$, where $Y$ symbolizes the corresponding consistent representation, **Consistency** for short, acting as the low-dimensional embedding shared across multiple views. The matrix $U^{(v)} \in \mathbb{R}^{m^{(v)} \times d}$ denotes the basis of the $v$-th view within the subspace $S$.

Drawing inspiration from the principles of consistency and supplementarity in MSL [7], we further break down the overall loss $\mu^{(v)}$ of the $v$-th view into two distinct components as follows:

*Definition 3.1.* **Consistency Loss & Supplementarity Loss**: The loss of mapping the feature matrix $X^{(v)} \in X$ of the $v$-th view to the unified space $S$ denoted by $\mu^{(v)}$ can be decomposed into the sum of consistency loss $\mu_c^{(v)}$ and supplementarity loss $\mu_s^{(v)}$.

Moreover, supplementarity is formulated as:

*Definition 3.2.* **Supplementarity (supplementary representation)**: The supplementarity loss while mapping the feature matrix $X^{(v)} \in X$ of the $v$-th view to a unified space $S$, denoted as $\mu_s^{(v)}$, is expressed as $\mu_s^{(v)} = U^{(v)} \hat{Y}^{(v)}$, where $\hat{Y}^{(v)}$ represents the supplementarity of $X^{(v)}$.

In essence, the subspace $S$ is reformulated as:

$$S = \{X^{(v)} \in X : X^{(v)} = U^{(v)}Y + \mu_c^{(v)} + U^{(v)}\hat{Y}^{(v)}\} \tag{3}$$

The term $\hat{Y}^{(v)}$ can be interpreted as the unique information from $X^{(v)}$ specific to the $v$-th view, not shared by other views. It represents the portion that cannot be uniformly expressed by the basis matrix $U^{(v)}$.

To tackle irrelevant variables, we introduce the following assumption:

ASSUMPTION 1. *The consistency loss for each view is hypothesized to be uniform and identical, implying $\mu_c^{(v)} \approx \mu_c^{(u)}, \forall \mathcal{V}^{(v)}, \mathcal{V}^{(u)} \in \mathcal{V} \wedge v \neq u$.*

Therefore, our attention can be singularly focused on supplementarity.

THEOREM 3.3. *Supplementarity Dominates the Mapping Loss: In Multi-view Subspace Learning (MSL), the differences in the mapping loss of two views to the subspace $S$ are primarily driven by their differences in supplementarity.*

PROOF. Consider the feature matrix $X^{(v)}$ of the $v$-th view and $X^{(u)}$ in the $u$-th view, without loss of generality, let $||X^{(v)}||$ represent the norm of $X(v)$. Follow the Definition 3.1, $||\mu^{(v)} - \mu^{(u)}|| = ||\mu_c^{(v)} + U^{(v)}\hat{Y}^{(v)} - \mu_c^{(u)} - U^{(u)}\hat{Y}^{(u)}||$. Given that $\mu_c^{(v)} \approx \mu_c^{(u)}$, $\forall \mathcal{V}^{(v)}, \mathcal{V}^{(u)} \in \mathcal{V} \wedge v \neq u$ based on Assumption 1, then $||\mu^{(v)} - \mu^{(u)}|| = ||U^{(v)}\hat{Y}^{(v)} - U^{(u)}\hat{Y}^{(u)}||$. As $U^{(v)}$ and $U^{(u)}$ are the basis matrices irrelevant to the loss computation, it follows that $||\mu^{(v)} - \mu^{(u)}||$ is primarily influenced by $||\hat{Y}^{(v)} - \hat{Y}^{(u)}||$. □

It is crucial to note that we can apply any arbitrary norm, such as the Frobenius norm [3], etc., as long as they satisfy the three

essential properties of a norm: Positivity, Scaling, and Triangle Inequality [15].

We now introduce the mathematical definition of the "Redundant Supplementarity" dilemma within the MSL.

*Definition 3.4.* **Redundant Supplementarity**: Assuming that $X^{(v)}$ adheres to Assumption 1, there exists a largest subset of $M$ views, denoted as $\mathcal{V}^M$, that exhibit similar supplementarity, i.e., $\hat{Y}^{(u)} \approx \hat{Y}^{(w)}$, for any $\mathcal{V}^{(u)}, \mathcal{V}^{(w)} \in \mathcal{V}^M \wedge u \neq w$, where $1 \leq M \leq V$. The issue of redundant supplementarity arises when the view-weights $\alpha^{(v)}$ allocated by MSL to these $M$ views are significantly higher than those allocated to the remaining $V - M$ views.

This scenario results in an excessive focus on these $M$ views with similar supplementarity, denoted as $\hat{Y}^M$, thereby biasing the learned shared representation towards $Y + \hat{Y}^M$. To investigate this matter, we introduce a function concerning view-weights:

*Definition 3.5.* **View-weight Difference Function** $f(M)$: $f(M) = \alpha^{(v)} - \alpha^{(u)}$, where $\mathcal{V}^{(v)} \in \mathcal{V}^M$ and $\mathcal{V}^{(u)} \notin \mathcal{V}^M$. The value of $f(M)$ fluctuates with changes in $M$.

Furthermore, we outline several key properties of $f(M)$:

THEOREM 3.6. *For View-weight Difference Function $f(M)$, the following properties hold: (1) Positivity: $f(M) \geq 0$. (2) Convergence: As $M \rightarrow V$, $f(M)$ will steadily converge to $\frac{1}{V}$.*

PROOF. Typical MSL assumes that all views contribute equally to consistency and supplementarity, setting $\alpha^{(v)}$ to $\frac{1}{V}$ [17, 29]. During iterative updates, $\alpha^{(v)}$ is adjusted to $\frac{\frac{1}{R^{(v)}}}{\sum_{u=1}^{V} \frac{1}{R^{(u)}}}$, where $R^{(v)} = \|X^{(v)} - U^{(v)} Y_{t=1}\|$.

The optimization objective function for updating $Y$ at the first iteration is:

$$\min \sum_{\mathcal{V}^{(v)} \in \mathcal{V}} \alpha^{(v)} \|X^{(v)} - U^{(v)} Y_{t=1}\|$$
$$= \min \frac{1}{V} \sum_{\mathcal{V}^{(v)} \in \mathcal{V}} \|U^{(v)} Y + U^{(v)} \hat{Y}^{(v)} - U^{(v)} Y_{t=1}\| \tag{4}$$

Recall the presence of $M$ views with similar supplementarity, denoted as $\hat{Y}^M$. Consequently, the objective function of the iterative updating can be expressed as:

$$\min \frac{1}{V} \sum_{\mathcal{V}^{(v)} \in \mathcal{V}^M} \|U^{(v)} Y + U^{(v)} \hat{Y}^M - U^{(v)} Y_{t=1}\|$$
$$+ \frac{1}{V} \sum_{\mathcal{V}^{(u)} \notin \mathcal{V}^M} \|U^{(u)} Y + U^{(u)} \hat{Y}^{(u)} - U^{(u)} Y_{t=1}\| \tag{5}$$

The $\alpha^{(v)}$ for $\mathcal{V}^{(v)} \in \mathcal{V}$ is iteratively updated as:

$$\alpha^{(v)} = \frac{\frac{1}{R^{(v)}}}{\sum_{\mathcal{V}^{(u)} \in \mathcal{V}^M} \frac{1}{R^{(u)}} + \sum_{\mathcal{V}^{(w)} \notin \mathcal{V}^M} \frac{1}{R^{(w)}}}$$
$$= \frac{\frac{1}{\|U^{(v)} Y + U^{(v)} \hat{Y}^{(v)} - U^{(v)} Y_{t=1}\|}}{R^M + R^{\bar{M}}}$$

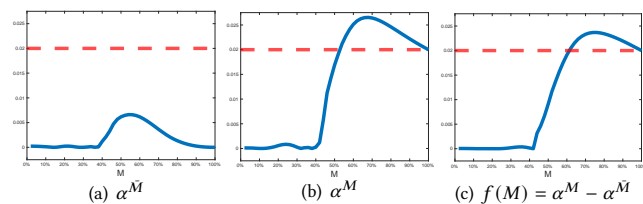

(a) $\alpha^M$     (b) $\alpha^M$     (c) $f(M) = \alpha^M - \alpha^{\bar{M}}$

**Figure 3: Illustration of $\alpha^M$, $\alpha^{\bar{M}}$, and $f(M)$. The Convergence Line ($\frac{1}{V} = \frac{1}{50} = 0.02$) is highlighted as red dashed.**

.

by substituting $R^{(v)} = \|U^{(v)} Y + U^{(v)} \hat{Y}^{(v)} - U^{(v)} Y_{t=1}\|$ and defining two components, $R^M$ and $R^{\bar{M}}$, as follows:

$$R^M = \sum_{\mathcal{V}^{(v)} \in \mathcal{V}^M} \frac{1}{\|U^{(v)} Y + U^{(v)} \hat{Y}^M - U^{(v)} Y_{t=1}\|}$$
$$R^{\bar{M}} = \sum_{\mathcal{V}^{(w)} \notin \mathcal{V}^M} \frac{1}{\|U^{(w)} Y + U^{(w)} \hat{Y}^{(w)} - U^{(w)} Y_{t=1}\|} \tag{6}$$

Moreover, $\alpha^{(v)}$ can be rewritten as:

$$\alpha^{(v)} = \begin{cases} \alpha^M = \dfrac{\frac{1}{\|U^{(v)} Y + U^{(v)} \hat{Y}^M - U^{(v)} Y_{t=1}\|}}{R^M + R^{\bar{M}}}, & \text{if } \mathcal{V}^{(v)} \in \mathcal{V}^M \\[3mm] \alpha^{\bar{M}} = \dfrac{\frac{1}{\|U^{(v)} Y + U^{(v)} \hat{Y}^{(v)} - U^{(v)} Y_{t=1}\|}}{R^M + R^{\bar{M}}}, & \text{otherwise} \end{cases} \tag{7}$$

Therefore, we further deduce:

$$f(M) = \alpha^M - \alpha^{\bar{M}} = \frac{1}{\sum_{\mathcal{V}^{(v)} \in \mathcal{V}} \frac{1}{R^{(v)}}} \cdot \left(\frac{R^{\bar{M}} - R^M}{R^{\bar{M}} R^M}\right). \tag{8}$$

The first term $\frac{1}{\sum_{\mathcal{V}^{(v)} \in \mathcal{V}} \frac{1}{R^{(v)}}}$ is always positive, and the change of $f(M)$ primarily depends on $R^{\bar{M}} - R^M$ in the second term. $R^{\bar{M}} \geq R^M$ since $\hat{Y}^M \geq \hat{Y}^{(w)}$ (as Eq. 6), thereby the Positivity holds.

Regarding the Convergence, according to Eq. 7, when $M \rightarrow V$, $Y_t$ approximates the average of consistency and supplementarity, $Y_t \rightarrow Y + \hat{Y}^M$. Therefore, $R^M \rightarrow \infty$ and $R^{\bar{M}} \rightarrow \frac{1}{\|U^{(v)} Y^{(v)} - U^{(v)} \hat{Y}^M\|}$, further we can derive $\alpha^M \rightarrow \frac{1}{V}$, $\alpha^{\bar{M}} \rightarrow 0$ and $f(M) \rightarrow \frac{1}{V}$. □

To examine the evolving pattern of $f(M)$, we randomly generated 50 views and initialized the first $M$ views as identical. Then, $M$ was incremented from 1 to $V$. In Figure 3, besides the $f(M)$ curve, the curves of $\alpha^M$ and $\alpha^{\bar{M}}$ are plotted. Observing Figure 3, it becomes apparent that with increasing $M$, the value of $\alpha^{\bar{M}}$ consistently remains below the convergence line ($\frac{1}{V}$: indicated by the red dashed line), whereas the value of $\alpha^M$ initially reaches a maximum before gradually decreasing and converging to the same line. The curve of $f(M)$ is notably influenced by that of $\alpha^M$. This trend underscores an unequal allocation of view-weights, a characteristic indicative of Redundant Supplementarity.

## 3.2 Equitable View-weight Allocation (EVA)

In the pursuit of EVA, our solution is straightforward: since redundant supplementarity emerges from sets of views sharing similar

supplementarity, views are partitioned as View Communities (VCs) by eliminating less similar links between them. For each VC, a VC-specific consistent representation, ensuring minimal yet adequate supplementarity, is created through an individual MSL process, given that MSL primarily emphasizes consistency in representation generation. EVA involves four key steps: (1) Compute the view-specific anchor graph $Z^{(v)}$ for each view. (2) Construct the View Supplementarity Graph (VSG). (3) Partition the VSG into View Communities (VCs). (4) Within each VC, apply MSL to generate a VC-specific consistency $Y_c$.

### 3.2.1 Compute view-specific anchor graph.
Traditional MVC methods often involve constructing a pairwise graph $A^{(v)} \in \mathbb{R}^{n \times n}$ for each view, where $n$ represents the number of data points. This representation typically decomposes into consistency and supplementarity components, expressed as $A^{(v)} = (Y + \hat{Y}^{(v)})$ [12], with $Y$ denoting the consistency and $\hat{Y}^{(v)}$ signifying the supplementarity of the $v$-th view. However, such a pairwise graph poses a significant computational burden due to its quadratic complexity.

In EVA-MVC, instead of focusing on $A^{(v)}$, we employ a more efficient anchor graph $Z^{(v)} \in \mathbb{R}^{p \times n}$ for each view, where $p$ (the number of anchors) $\ll n$. When $p = n$, $Z^{(v)}$ is equivalent to $A^{(v)}$.

To mitigate the impact of misaligned anchors on self-representation computation, we first concatenate the multi-views into a single matrix, denoted as $X_t \in \mathbb{R}^{\sum m^{(v)} \times n}$. Subsequently, we perform $k$-means on $X_t$ to generate $p$ anchors, denoted as $P \in \mathbb{R}^{\sum m^{(v)} \times p}$. Utilizing these anchors, the view-specific anchor graph is computed as:

$$Z_{ij}^{(v)} = \exp(-\frac{\left\|X_{:j}^{(v)} - P_{:j}^{(v)}\right\|_2^2}{2\sigma^2}), \tag{9}$$

where $Z^{(v)} \in \mathbb{R}^{p \times n}$, and $P^{(v)}$ represents the anchors specific to each view, derived from $P$.

### 3.2.2 Construct View Supplementarity Graph (VSG).
To quantify the similarity in supplementarity between any two views, we utilize the Frobenius norm [3] for evaluation, expressed as follows:

$$\begin{aligned} G_{ij} &= \frac{1}{1 + \|Z^{(i)} - Z^{(j)}\|_F^2} = \frac{1}{1 + \|Y + \hat{Y}^{(i)} - (Y + \hat{Y}^{(j)})\|_F^2} \\ &= \frac{1}{1 + \|\hat{Y}^{(i)} - \hat{Y}^{(j)}\|_F^2} \end{aligned} \tag{10}$$

Eq. 10 uses a reciprocal kernel based on the Frobenius norm. The VSG $G \in \mathbb{R}^{V \times V}$ is a complete graph, where views act as the nodes and edges denote the similarity in supplementarity. Eq. 10 also exemplifies Theorem 3.6, where $G_{ij}$ is primarily influenced by the Frobenius norm in supplementarity, $\|\hat{Y}^{(i)} - \hat{Y}^{(j)}\|_F^2$. Additionally, we define $Z^{(v)} = (Y + \hat{Y}^{(v)})$ by regarding $Z^{(v)}$ as a special form of $A^{(v)}$. The proof of equivalence between the anchor graph $Z^{(v)}$ and pairwise graph $A^{(v)}$ in VSG construction is provided in the Appendix. Further, we acknowledge the existence of other matrix norms or kernels, which are beyond the scope of this study.

### 3.2.3 Partition VSG.
Following the construction of the VSG, the subsequent step involves partitioning it into distinct View Communities (VCs). Drawing on density-based partitioning methods [33],

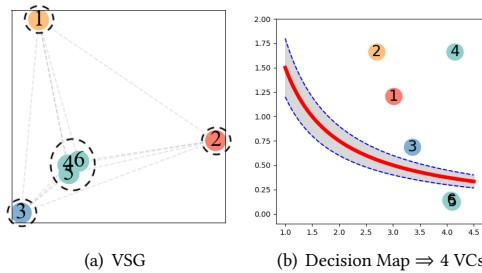

(a) VSG      (b) Decision Map $\Rightarrow$ 4 VCs

**Figure 4: View Supplimentarity Graph (VSG) Plot and Decision Map of Caltech101-7 dataset. The decision boundary is uniformly set as $y = \frac{\tau}{x}$, where $\tau = 1.5 \pm 0.3$ for all datasets.**

we establish the supplementarity-based density and dependent distance metrics for each view as outlined below:

$$\rho_i = \sum_j G_{ij}, \quad \delta_i = \min_{j:\rho_j > \rho_i} (\|Z^{(i)} - Z^{(j)}\|_F^2). \tag{11}$$

where $\rho_i$ denotes the density of the $i$-th view, calculated by aggregating its supplementarity relevance. The dependent distance of the $i$-th view, $\delta_i$, is defined as the Frobenius norm distance from the $i$-th view to its nearest denser view (where $\rho_j > \rho_i$).

Our aim is to find views that exhibit significant influence (high density $\rho_i$) and broad coverage (large dependent distance $\delta_i$) as the modes of View Communities (VCs). Thus, a view is designated as a VC mode if:

$$\rho_i \cdot \delta_i > \tau \tag{12}$$

where $\tau$ is a predefined threshold. This identification process can be visually depicted through a decision map, with the axes representing density and dependent distance. As shown in Figure 4 (b), views located closer to the top-right corner in this map are more likely to act as VC modes. The threshold $\tau$, depicted as a gray region, functions as the decision boundary, identifying views falling within its top-right part as the VC modes. The remaining views are assigned to the closest mode in dependent distance.

The obtained View Communities (VCs) can be expressed as:

$$\mathcal{VC} = \{\mathcal{VC}^{(1)}, \mathcal{VC}^{(2)}, \ldots, \mathcal{VC}^{(C)}\}, \tag{13}$$

where $\cup_{c=1}^C \mathcal{VC}^{(c)} = \mathcal{V}$, $C$ is the number of VCs, $C^{(c)}$ denotes the number of views in the $c$-th VC, $\mathcal{VC}^{(c)} \cap \mathcal{VC}^{(d)} = \emptyset$. Meanwhile, for obtained VCs, their VC-specific Consistencies (consistent representation) $\{Y_1, \cdots, Y_C\}$ are generated. $Y_c$, the consistency of the $c$-th VC is computed as follows [17]:

$$\begin{aligned} \min_{\alpha, H_c^{(v)}, Y_c} \sum_{\mathcal{V}^{(v)} \in \mathcal{VC}^{(c)}} \frac{(\alpha^{(v)})^2}{2} \|X^{(v)} - U_c^{(v)} Y_c\|_F^2, \\ s.t. \ (\alpha^{(v)})^T \mathbf{1} = 1, \alpha^{(v)} \geq 0, Y_c Y_c^T = I \end{aligned} \tag{14}$$

We now prove that EVA can avoid redundant supplementarity.

**Theorem 3.7.** *Assuming the multi-view dataset $X$ is governed by Assumption 1 and views are partitioned into $C$ View Communities, denoted as $\mathcal{VC}$, based on their pairwise supplementarity similarities. There exists a largest View Community, referenced as $\mathcal{VC}^M$, with a*

---

**Algorithm 1:** Pseudocode of EVA-MVC

**Input:** $\left\{ X^{(v)} \in \mathbb{R}^{m^{(v)} \times n} \right\}_{v=1}^{V}, \tau, r, l, \lambda(10^{-5}), k.$

/* EVA module: Equitable View-weight Allocation    */

1  $Z^{(v)} \leftarrow$ For each view $X^{(v)}$, build the anchor graph by Eq. 9 .

2  $G \leftarrow$ Construct the View Supplementarity Graph (VSG) by Eq. 10.

3  $\mathcal{VC} \leftarrow$ Partition the VSG into View Communities (VCs).

/* A process of MSL is performed within each VC    */

4  $Y_c \leftarrow$ For each $\mathcal{VC}^{(c)}$, learn its VC-specific consistency by Eq. 14.

/* Clustering module: Clustering by any MVC method    */

5  Perform an arbitrary MVC method on $\{Y_1, \cdots, Y_C\}$.

**Output:** $k$ clusters of $\left\{ X^{(v)} \in \mathbb{R}^{m^{(v)} \times n} \right\}_{v=1}^{V}$.

---

size $M$, where $M = \max(C^{(c)})$. As $M \to V$, the view-weight of the $v$-th view $\alpha^{(v)}$, learned through a standard MSL, will converge towards $\frac{1}{C^{(c)}}$, becoming irrelevant to $M$, for $\mathcal{V}^{(v)} \in \mathcal{VC}^{(c)} \wedge \mathcal{V}^{(v)} \notin \mathcal{VC}^{M}$.

The proof of Theorem 3.7 follows a similar structure to that of Theorem 3.6, with its detailed exposition available in the Appendix. Besides, the VSG partition method described above is independent of EVA and can be replaced by other density-based partition algorithms, if desired. Further details regarding an alternative VSG partition algorithm can be found in the Appendix.

### 3.3 Integrate EVA with Arbitrary MVC Method

As outlined in Algorithm 1, our proposed framework is structured around two modules: EVA and Clustering. Within the EVA module, the initial step involves partitioning the views into VCs $\mathcal{VC}$ (lines 1-3). Subsequently, within each VC, its consistency is computed through an MSL process [17] (line 4). These consistent representations, in the clustering module, function as the multi-view feature matrices and are fused using a standard MVC method (line 5).

Viewed from the Information Fusion [20] aspect, EVA-MVC embodies a two-stage fusion architecture with EVA representing Early Fusion and MVC representing Late Fusion. The parameter $\tau$ is instrumental in balancing the interplay between these two fusion stages. As $\tau \to 0$, each view constitutes a community, rendering EVA inactive, thereby aligning our framework with the MVC method executed in the clustering module. Conversely, as $\tau \to \infty$, all views amalgamate into a singular community, aligning our framework with Early Fusion MSL [8].

### 3.4 Complexity & Convergence Analysis

**Time complexity.** (1) EVA module: The time complexity is $O(tkmn + V^2)$, where $t$ is the number of iterations of $k$-means. (2) Clustering module: The time complexity hinges on the chosen MVC method. As detailed in Table 1, considering the method OMSC [8], which requires a computational cost of $O(n)$. Thus, the time complexity of our framework is nearly linear to the number of data points $O(n)$.

**Space complexity.** The space complexity primarily revolves around storing matrices $\{U_c^{(v)} \in \mathbb{R}^{m^{(v)} \times k}\}_{v=1}^{V}$ and $\{Y_c \in \mathbb{R}^{k \times n}\}_{c=1}^{C}$. Thus, the space complexity of EVA-MVC is nearly linear to the number of data points, $O(n)$ as well.

**Convergence.** There is no convergence problem in EVA module, while regarding the MVC module, it depends on the chosen MVC

**Table 2: Dataset statistics**

| Datasets | #Objects | #Views | #Classes | #View Dimensions |
|---|---|---|---|---|
| uci-digit | 2000 | 3 | 10 | 216, 76, 64 |
| 3Sources | 169 | 3 | 6 | 3560, 3631, 3068 |
| Caltech101-7 | 1474 | 6 | 4 | 48, 40, 254, 1984, 512, 928 |
| CiteSeer | 3312 | 2 | 6 | 3312, 3703 |
| Animal | 11673 | 4 | 20 | 2689, 2000, 2001, 2000 |
| CIFAR-10 | 50000 | 3 | 10 | 512, 2048, 1024 |
| YTF10 | 38654 | 4 | 10 | 944, 576, 512, 640 |
| YTF20 | 63896 | 4 | 20 | 944, 576, 512, 640 |
| YTF50 | 126054 | 4 | 50 | 944, 576, 512, 640 |
| YTF100 | 195537 | 4 | 50 | 944, 576, 512, 640 |

method. Given the versatility of EVA, we choose a well-known MSL method, OMSC [8], as the default of EVA-MVC. Inspired by work [2] and [44], we present a comprehensive mathematical proof for the convergence of EVA-MVC in the Appendix.

## 4 EVALUATION

### 4.1 Datasets and Baselines

**Datasets.** We evaluate the clustering effect on the following datasets: (1) **Uci-digit.**[1] Ten classes of handwritten digits, with 200 examples per class. (2) **3Source.**[2] 169 news articles from BBC, Reuters, and Guardian. (3) **Caltech101-7.**[3] Seven categories from the Caltech101 dataset. (4) **CiteSeer.** 3,312 scientific publications classified into six categories. (5) **Animal.** 50 animal species described by four features. (6) **CIFAR-10.** A subset of labeled images including 10 categories. (7) **YTF10, YTF20, YTF50, and YTF100**[4]. These are four versions of the YouTube-Faces (YTF) dataset. The purpose of testing different versions (of different data sizes) of these large-scale datasets is to better evaluate the MVC algorithms with different levels of scalability. Please refer to Table 2 for details.

**Baselines**. OMSC [8] is used as the default MVC method for EVA-MVC. For comprehensive evaluation, we selected baselines from each category of MVC (as in Table 1). Only the methods with available source code were evaluated: (1) **LF-LAM** [43]: A late-stage fusion multi-view clustering method. (2) **FPMVS** [44]: Constructs a low-rank graph without hyperparameters. (3) **SMVSC** [35]: Proposes a unified framework of anchor learning and graph construction. (4) **OMSC** [8]: Enhances anchor representation and clustering by a unified framework. (5) **FMVACC** [42]: Finds anchor correspondences using feature and structure information. (6) **AWMVC** [39]: Merges coefficient matrices from base matrices to form an optimal consensus matrix. (7) **FastMICE** [20]: Introduces random-view groups to capture multi-functional view relationships. (8) **EMVGC-LG** [47]: An anchor-based framework preserving local and global structures. (9) **CMVC** [57]: Proposes an adaptive Cluster-wise Anchor learning method.

---

[1]http://archive.ics.uci.edu/ml/datasets/Multiple+Features
[2]http://mlg.ucd.ie/datasets/3sources.html
[3]http://www.vision.caltech.edu/Image_Datasets/Caltech101/
[4]https://www.cs.tau.ac.il/~wolf/ytfaces/

**Table 3: Comparison results.**

| Datasets | Metric | LF-LAM | FPMVS | MSGL | SMVSC | OMSC | FMVACC | AWMVC | FastMICE | EMVGC-LG | CMVC | EVA-MVC |
|---|---|---|---|---|---|---|---|---|---|---|---|---|
| uci-digit | ACC | 0.9005±0.014 | 0.8265±0.000 | 0.7380±0.029 | 0.8055±0.000 | 0.7325±0.000 | 0.7716±0.062 | 0.7749±0.015 | 0.8070±0.039 | 0.8889±0.025 | 0.9205±0.000 | **0.9270±0.000** |
| | NMI | 0.8186±0.017 | 0.8124±0.000 | 0.7417±0.016 | 0.7557±0.000 | 0.7490±0.000 | 0.7339±0.030 | 0.7427±0.009 | 0.8185±0.028 | 0.8380±0.017 | 0.8576±0.000 | **0.8576±0.000** |
| | Purity | 0.9005±0.014 | 0.8270±0.000 | 0.8120±0.027 | 0.8055±0.000 | 0.7395±0.000 | 0.7933±0.047 | 0.7935±0.014 | 0.8603±0.039 | 0.8932±0.023 | 0.9205±0.000 | **0.9270±0.000** |
| | Fscore | 0.8160±0.022 | 0.7733±0.000 | 0.6606±0.022 | 0.7003±0.000 | 0.6848±0.000 | 0.6923±0.043 | 0.6987±0.013 | 0.8054±0.040 | 0.8182±0.026 | 0.8517±0.000 | **0.8621±0.000** |
| 3Source | ACC | 0.5207±0.009 | 0.4201±0.000 | 0.3353±0.012 | 0.3786±0.000 | 0.3372±0.000 | 0.4923±0.044 | 0.6450±0.010 | 0.5041±0.054 | 0.5321±0.050 | 0.7396±0.000 | **0.7692±0.000** |
| | NMI | 0.5110±0.010 | 0.1578±0.000 | 0.0630±0.013 | 0.1117±0.000 | 0.1271±0.000 | 0.3792±0.051 | 0.5419±0.007 | 0.4135±0.053 | 0.4778±0.048 | 0.6566±0.000 | **0.7408±0.000** |
| | Purity | 0.7337±0.015 | 0.5325±0.000 | 0.6331±0.135 | 0.4378±0.000 | 0.4378±0.000 | 0.6047±0.040 | 0.7343±0.006 | 0.6177±0.019 | 0.6608±0.038 | 0.7870±0.000 | **0.8579±0.000** |
| | Fscore | 0.4822±0.015 | 0.3391±0.000 | 0.3713±0.026 | 0.2914±0.000 | 0.2530±0.000 | 0.4100±0.043 | 0.5633±0.009 | 0.4022±0.054 | 0.4590±0.047 | 0.7079±0.000 | **0.7531±0.000** |
| Caltech101-7 | ACC | 0.4322±0.027 | 0.6872±0.000 | 0.6282±0.088 | 0.7014±0.000 | 0.6614±0.000 | 0.4105±0.017 | 0.3693±0.008 | 0.5362±0.011 | 0.3708±0.017 | 0.5039±0.000 | **0.8853±0.000** |
| | NMI | 0.5091±0.030 | 0.5055±0.000 | 0.4448±0.114 | 0.5633±0.000 | 0.5567±0.000 | 0.3939±0.017 | 0.4957±0.009 | 0.5778±0.013 | 0.4870±0.015 | 0.5647±0.000 | **0.6983±0.000** |
| | Purity | 0.8541±0.020 | 0.8086±0.000 | 0.7069±0.067 | 0.8656±0.000 | 0.8588±0.000 | 0.7734±0.022 | 0.8348±0.005 | 0.6160±0.018 | 0.8226±0.007 | 0.8559±0.000 | **0.9084±0.000** |
| | Fscore | 0.4529±0.008 | 0.6728±0.000 | 0.5956±0.063 | 0.6809±0.000 | 0.6503±0.000 | 0.4114±0.018 | 0.4364±0.007 | 0.5714±0.014 | 0.4135±0.012 | 0.5485±0.000 | **0.8615±0.000** |
| CiteSeer | ACC | 0.3897±0.022 | 0.3867±0.000 | 0.2137±0.005 | 0.3734±0.000 | 0.3867±0.000 | 0.4480±0.074 | 0.4300±0.055 | 0.4355±0.023 | 0.3969±0.091 | 0.5284±0.000 | **0.5845±0.000** |
| | NMI | 0.1604±0.028 | 0.1439±0.000 | 0.0109±0.004 | 0.1486±0.000 | 0.1472±0.000 | 0.2280±0.060 | 0.2332±0.014 | 0.2195±0.025 | 0.2272±0.074 | 0.2607±0.000 | **0.3331±0.000** |
| | Purity | 0.4142±0.021 | 0.4060±0.000 | 0.2171±0.005 | 0.4018±0.000 | 0.4344±0.000 | 0.4849±0.077 | 0.4943±0.037 | 0.4905±0.023 | 0.4172±0.086 | 0.5501±0.000 | **0.6237±0.000** |
| | Fscore | 0.2869±0.017 | 0.2908±0.000 | 0.2965±0.012 | 0.2853±0.000 | 0.2916±0.000 | 0.3340±0.051 | 0.3403±0.024 | 0.3237±0.017 | 0.3240±0.029 | 0.3769±0.000 | **0.4451±0.000** |
| Animal | ACC | N/A | **0.2026±0.000** | 0.1350±0.005 | 0.1740±0.000 | 0.1804±0.000 | 0.1334±0.002 | 0.1494±0.004 | 0.1641±0.003 | 0.1765±0.008 | 0.1727±0.000 | 0.1883±0.000 |
| | NMI | N/A | **0.1596±0.000** | 0.0935±0.004 | 0.1444±0.000 | 0.1434±0.000 | 0.0896±0.001 | 0.1246±0.003 | 0.1299±0.004 | 0.1413±0.004 | 0.1541±0.000 | 0.1503±0.000 |
| | Purity | N/A | 0.2124±0.000 | 0.1742±0.006 | 0.2045±0.000 | 0.2050±0.000 | 0.1672±0.002 | 0.1869±0.005 | 0.1858±0.005 | 0.2068±0.009 | 0.2190±0.000 | **0.2207±0.000** |
| | Fscore | N/A | **0.1466±0.000** | 0.0978±0.003 | 0.1045±0.000 | 0.1314±0.000 | 0.0825±0.001 | 0.0975±0.001 | 0.1020±0.001 | 0.1139±0.003 | 0.1099±0.000 | 0.1117±0.000 |
| CIFAR-10 | ACC | N/A | 0.9898±0.000 | 0.9314±0.028 | 0.9882±0.000 | 0.9885±0.000 | 0.9535±0.049 | 0.9282±0.090 | 0.9500±0.056 | 0.9154±0.045 | 0.9931±0.000 | **0.9944±0.000** |
| | NMI | N/A | 0.9729±0.000 | 0.8843±0.034 | 0.9690±0.000 | 0.9697±0.000 | 0.9365±0.017 | 0.9112±0.032 | 0.9625±0.018 | 0.9178±0.018 | 0.9811±0.000 | **0.9841±0.000** |
| | Purity | N/A | 0.9898±0.000 | 0.9314±0.027 | 0.9882±0.000 | 0.9885±0.000 | 0.9541±0.048 | 0.9394±0.064 | 0.9899±0.001 | 0.9214±0.037 | 0.9931±0.000 | **0.9944±0.000** |
| | Fscore | N/A | 0.9800±0.000 | 0.8763±0.023 | 0.9767±0.000 | 0.9773±0.000 | 0.9360±0.043 | 0.9094±0.067 | 0.9463±0.050 | 0.8995±0.041 | 0.9864±0.000 | **0.9890±0.000** |
| YTF100 | ACC | N/A | 0.5293±0.000 | 0.4340±0.035 | 0.5906±0.000 | 0.6651±0.000 | 0.6344±0.002 | 0.6283±0.010 | 0.6683±0.016 | 0.6195±0.014 | 0.6652±0.000 | **0.7531±0.000** |
| | NMI | N/A | 0.7532±0.000 | 0.6342±0.046 | 0.7991±0.000 | 0.8337±0.000 | 0.8190±0.004 | 0.8304±0.002 | 0.8309±0.069 | 0.8247±0.005 | 0.8318±0.000 | **0.8492±0.000** |
| | Purity | N/A | 0.5446±0.000 | 0.6100±0.038 | 0.6103±0.000 | 0.7141±0.000 | 0.6659±0.020 | 0.7212±0.007 | 0.7359±0.014 | 0.7163±0.008 | 0.7375±0.000 | **0.8000±0.000** |
| | Fscore | N/A | 0.3541±0.000 | 0.1562±0.033 | 0.5035±0.000 | 0.5846±0.000 | 0.5765±0.026 | 0.5297±0.007 | 0.6000±0.022 | 0.5147±0.006 | 0.5850±0.000 | **0.7043±0.000** |

**Experiment Setup**. To evaluate the effectiveness of the proposed clustering method, we employed four commonly used performance measures: **ACC** (Accuracy), **NMI** (Normalized Mutual Information), **Purity**, and **F-score** [40]. We used open-source methods and configured their respective parameters according to the specifications outlined in their papers. For fair comparison, all methods were run 10 times and and the average results were reported. For EVA-MVC, we varies the following parameters: $\tau = 1.5$, $r \in \{k, 2k, 3k\}$, $l \in \{k, 2k, 3k\}$, and $\lambda \in 0.00001$, where $k$ denotes the number of clusters. The best results by varying these parameters are reported. Only a subset of experiments is presented in this section; the comprehensive set is available in the Appendix.

## 4.2 Comparison Results

**Clustering Performance**. Table 3 compares clustering performance on seven benchmark datasets, with some methods showing zero variance due to intentional initialization. (1) Compared with MSL methods (i.e., FPMVS, SMVSC, OMSC, AWMVC) suffering from Redundant Supplementarity, our algorithm alleviates the problem of poor subspace quality caused by fusing views with significantly different anchor graphs. Across nine datasets, our approach achieves higher ACC by 3.81%, 12.42%, 18.39%,15.45%,0.46%

0.34%, 3.30%, 7.66%, and 8.80%, respectively, demonstrating the superiority of partitioning view communities. (2) Compared with the methods that perform MSL independently in each view (i.e., LF-MVC-LAM and FMVACC), our algorithm effectively exploits rich and supplementarity from multiple views. As a result, the ACC on nine datasets is higher by 2.65%, 24.85%, 45.31%,13.65%,4.09%, 7.13%, 8.93%, 12.14%, and 8.80%, respectively. (3) Compared with the random fusion method (i.e., FastMICE), our EVA is more concise and mitigates the adverse effects of redundant and incompatible view communities on MSL. Consequently, the ACC on the nine datasets increased by 8.00%, 26.51%, 34.91%,14.90%, 4.44%, 1.51%, 9.79%, 9.97%, and 8.48%, respectively.

**Running Time**. In the runtime comparison on large-scale datasets (logarithmic scale on the y-axis), Figure 5 shows that EVA-MVC ranks third. While slightly slower than AWMVC, EVA-MVC significantly outperforms it in clustering performance. Although Fast-MICE is faster, its random view group partitioning leads to poorer clustering results. In contrast, EVA-MVC balances running time and clustering performance by partitioning views into VCs. Thus, EVA-MVC demonstrates superior overall performance.

**Parameter Analysis**. We conducted parameter analysis on three major parameters of EVA-MVC: $\tau$, $r$, and $l$, where ACC and

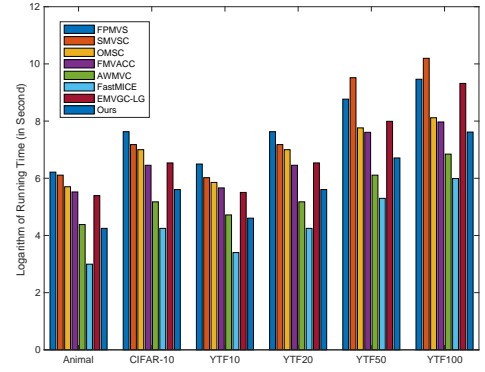

Figure 5: Running time comparison

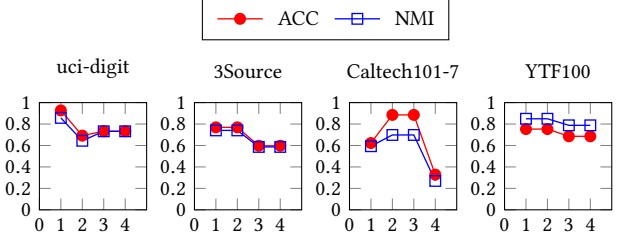

Figure 6: Parameter analysis on $\tau$ ($x$-axis)

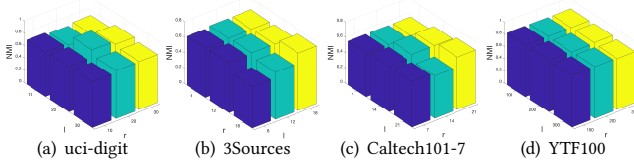

(a) uci-digit  (b) 3Sources  (c) Caltech101-7  (d) YTF100

Figure 7: Parameter analysis on $l$ ($x$-axis) and $r$ ($y$-axis)

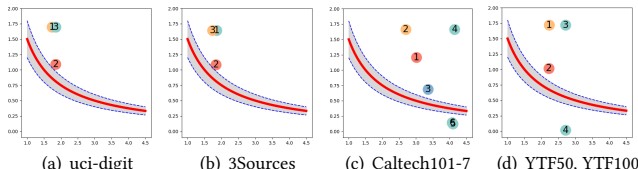

(a) uci-digit  (b) 3Sources  (c) Caltech101-7  (d) YTF50, YTF100

Figure 8: Decision Maps Plots. The decision boundary is uniformly set as $y = \frac{\tau}{x}$, where $\tau = 1.5 \pm 0.3$.

NMI are used as metrics. Figure 6 exhibits that EVA-MSC achieves the optimal performance when decision boundary $\tau$ falls within the range of $1.5 \pm 0.3$.

Figure 7 illustrates a three-dimensional histogram to assess the influence of parameters $r$ and $l$ on the clustering outcomes. Our analysis reveals that variations in parameters $r$ and $l$ have minimal effect on the clustering results when they fall within the range $[k, 3k]$. Hence, we recommend setting $r$ and $l$ within this range.

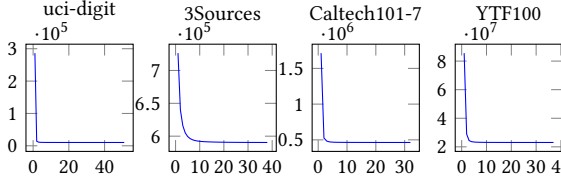

Figure 9: Convergence analysis ($x$-axis: no. of Iterations)

Table 4: The ablation study of EVA-MVC. (Metric: NMI)

| Datasets | LVA-MVC | RVE-MVC | NVA-MVC | EVA-MVC |
|---|---|---|---|---|
| uci-digit | 0.6456 | 0.7064 | 0.8124 | **0.8285** |
| 3Source | 0.5867 | 0.5163 | 0.1578 | **0.7408** |
| Caltech101-7 | 0.3638 | 0.4056 | 0.5055 | **0.6983** |
| YTF10 | 0.7689 | 0.7581 | 0.7740 | **0.8351** |
| YTF20 | 0.6543 | 0.7772 | 0.7740 | **0.8146** |
| YTF50 | 0.7975 | 0.5565 | 0.8364 | **0.8487** |
| YTF100 | 0.7539 | 0.5724 | 0.7532 | **0.8492** |

**Convergence Evaluation**. Figure 9 provides convergence curves of EVA-MVC, all of which indicate a sharp convergence within the first 10 iterations, followed by a stabilization in subsequent iterations. It underscores the efficiency of EVA-MVC in achieving clustering results with minimal iterations.

**Ablation Study**. To assess EVA-MVC, we conducted an ablation study by generating three variants: LVA-MVC, RVA-MVC, and NVA-MVC. The remaining aspects are kept identical to EVA-MVC. LVA-MVC: Views are partitioned using the opposite partition criteria to that of EVA-MVC, keeping views with the lowest supplementarity within a VC. RVA-MVC: Views are randomly and repetitively assigned to VCs. NVA-MVC: Does not partition views and directly applies early fusion to all views.

The ablation study results presented in Table 4 lead to several conclusions: (1) The quality of VC-specific consistency is crucial for the clustering module, significantly benefiting from supplementarity-driven partitioning. (2) Randomly assigning diminishes the consistency quality due to the uncertainty associated with VCs. (3) Without EVA, our framework reverts to the conventional Early Fusion MVC approach. These findings illustrate that the proposed EVA effectively addresses the issue of Redundant Supplementarity, thereby enhancing subsequent tasks such as clustering.

## 5 CONCLUSION

In this paper, we defined Redundant Supplementarity within MSL and introduced the EVA-MVC framework to address it by ensuring Equitable View-weight Allocation (EVA) through organizing similar views into VC groups before MSL. Theoretical analyses and extensive experiments validate that EVA-MVC effectively resolves this issue and enhances overall MVC quality. This framework marks a significant stride in boosting the robustness and accuracy of MSL, leading to more equitable and efficient methods in this domain. Our future research will delve into exploring Redundant Supplementarity in broader MVC contexts.

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

# 6 APPENDIX

## 6.1 Equivalence between Anchor Graph-based EVA and Pairwise Graph-based EVA

THEOREM 6.1. *Equivalence of Anchor Graph and Pairwise Graph*: In assessing the difference in supplementarity between two views, the anchor graph, $Z^{(v)}$, is equivalent to the pairwise graph, $A^{(v)}$.

PROOF. To establish the equivalence between the anchor graph $Z^{(v)}$ and the pairwise graph $A^{(v)}$, it is essential to demonstrate that $Z^{(v)}$ processes the properties of a matrix norm, just like $A^{(v)}$, including Positivity, Scaling, and the Triangle Inequality.

Firstly, we prove the Positivity of $Z^{(v)}$:

$$Z^{(v)} = E \cdot A^{(v)}, \tag{15}$$

where $E$ is a permutation matrix selecting $p$ columns from the $A^{(v)}$ matrix to construct $Z^{(v)}$. Consequently,

$$\|A^{(i)} - A^{(j)}\| = \|E \cdot (Z^{(i)} - Z^{(j)})\| \le \|E\| \cdot \|Z^{(i)} - Z^{(j)}\|. \tag{16}$$

Given that $\|A^{(i)} - A^{(j)}\| \ge 0$ and $\|E\| \ge 0$, we can conclude that $\|Z^{(i)} - Z^{(j)}\| \ge 0$, satisfying the Positivity criterion.

The Scaling ($\|\alpha Z^{(v)}\| = |\alpha| \cdot \|Z^{(v)}\|$) and Triangle Inequality ($\|Z^{(i)} + Z^{(j)}\| \le \|Z^{(i)}\| + \|Z^{(j)}\|$) properties of $Z^{(v)}$ can be similarly deduced. □

## 6.2 Proof of Theorem 3.7

PROOF. EVA partitions views into VCs based on pairwise supplementarity similarities, i.e., $\forall \mathcal{V}^{(v)}, \mathcal{V}^{(u)} \in \mathcal{VC}^{(c)} \wedge v \ne u$, $\hat{Y}^{(v)} \approx \hat{Y}^{(u)}$.

During iterative updates, for the $v$-th view in the $c$-th VC, its view-weight $\alpha^{(v)}$ is allocated as $\frac{\frac{1}{s^{(v)}}}{\sum_{\mathcal{V}^{(u)} \in \mathcal{VC}^{(c)}} s^{(u)}}$, where $s^{(v)} = \|X^{(v)} - U_c^{(v)}(Y_c)_{t=1}\|$.

At the initial update, the optimization objective function for updating $(Y_c)_{t=1}$ is:

$$
\begin{aligned}
&\min \sum_{\mathcal{V}^{(u)} \in \mathcal{VC}^{(c)}} \alpha^{(u)} \|X^{(u)} - U_c^{(u)}(Y_c)_{t=1}\| \\
&= \min \frac{1}{C^{(c)}} \sum_{\mathcal{V}^{(u)} \in \mathcal{VC}^{(c)}} \|U^{(u)}Y + U^{(u)}\hat{Y}^{(u)} - U_c^{(u)}(Y_c)_{t=1}\|
\end{aligned}
\tag{17}
$$

Given $C^{(c)}$ similar views in $\mathcal{VC}^{(c)}$ including $\mathcal{V}^{(v)}$, their supplementarity can be uniformly expressed as $\hat{Y}^{C^{(c)}}$. Consequently, the updated objective function can be expressed as:

$$
\begin{aligned}
&\min \sum_{\mathcal{V}^{(u)} \in \mathcal{VC}^{(c)}} \alpha^{(u)} \|X^{(u)} - U_c^{(u)}(Y_c)_{t=1}\| \\
&= \min \frac{1}{C^{(c)}} \sum_{\mathcal{V}^{(u)} \in \mathcal{VC}^{(c)}} \|U^{(u)}Y + U^{(u)}\hat{Y}^{C^{(c)}} - U_c^{(u)}(Y_c)_{t=1}\|
\end{aligned}
$$

The view-weight $\alpha^{(v)}$, at each iterative update can be given as:

$$\alpha^{(v)} = \frac{\frac{1}{s^{(v)}}}{\sum_{\mathcal{V}^{(u)} \in \mathcal{VC}^{(c)}} \frac{1}{s^{(u)}}} \tag{18}$$

By substituting $s^{(v)} = \|U^{(v)}Y + U^{(v)}\hat{Y}^{C^{(c)}} - U_c^{(v)}(Y_c)_{t=1}\|$, we derive:

$$\alpha^{(v)} = \frac{\frac{1}{\|U^{(v)}Y + U^{(v)}\hat{Y}^{C^{(c)}} - U_c^{(v)}(Y_c)_{t=1}\|}}{\sum_{\mathcal{V}^{(u)} \in \mathcal{VC}^{(c)}} \frac{1}{\|U^{(u)}Y + U^{(u)}\hat{Y}^{C^{(c)}} - U_c^{(u)}(Y_c)_{t=1}\|}} \approx \frac{1}{C^{(c)}}$$

Hence, $\alpha^{(v)}$ converges towards $\frac{1}{C^{(c)}}$, irrespective of $M$. □

## 6.3 Alternative View Supplimentarity Graph (VSG) Partition Algorithm

In this section, we introduce an alternative VSG partitioning algorithm that can serve as a replacement for the one outlined in the main text (Line 3 of the Algorithm 1).

Drawing inspiration from cluster tree-based clustering algorithms [4], we define $\theta_K(v) := \inf\{\theta > 0 : |B(v, \theta) \cap Z_{[V]}| \ge K\}$, i.e., as the distance from $v$ to its $K$-th nearest neighbor, and according to [22], $1/\theta_K(v)$ can be viewed as the density of node $v$. In our scenario, $\theta_K(v) = 1/G_{vu}$ (as Eq. 10 in main text), where $u$ is the $K$-th most similar view of $v$ in terms of supplementarity, and $G_{vu}$ denotes their supplementarity.

The alternative VSG partition algorithm, outlined in Algorithm 2, proceeds as follows:

(1) Initialize all nodes as disconnected and compute their densities, $\theta_K(v)$.

(2) Traverse the nodes in descending order of density, i.e., $1/\theta_K(v)$ (Line 2);

(3) For each node $v$ with density higher than the threshold $\theta$, check if there exist other nodes whose distance to $v$ is smaller than their densities. If so, connect and merge these nodes into the same community. (Lines 3-4).

(4) Decrease the threshold $\theta$ and repeat steps 3-4 until $\theta$ reaches its minimum.

(5) The resulting connected nodes form View Communities (Output).

---

**Algorithm 2:** Alternative VSG Partition Algorithm

**Input:** $VSG, K = V - 1, \gamma = 0.3$.

1 For each $Z^{(v)}$ computes $\theta_K(v)$.
2 **for** $\theta$ grows from $0$ to $\max(\theta_K(v))$: **do**
3      Construct a graph $CG_\theta^{NN}$ with nodes $\{N^{(i)} : (\theta_K(i) \leq \theta)\}$.
4      Include edge $(i, j)$ if $G_{ij} \leq \gamma \min(\theta_K(i), \theta_K(j))$.
5 **end**
6 Compute the connected components of $CG_\theta^{NN}$.

**Output:** $G$ communities of $N_{[V]}$.

---

This demonstrates that our EVA module is independent of the embedded VSG partition algorithm, as the VSG partitioning algorithm described in the main text can be substituted with Algorithm 2. Both algorithms share a common intuition:

(1) The view density (i.e., view degree) is used to estimate the impact of a view (node) by computing its "centrality" across multiple views. A view with a higher density value is more likely to serve as the View Community Mode (central view).

(2) Views with lower impact values are either dominated by nearby VC modes due to their proximity (i.e., distance) or serve as new VC Modes if they are far away from other Modes.

In summary, any density-based clustering algorithms that capture both the density and distance correlations of views can be seamlessly integrated into EVA after thorough analysis.

## 6.4 Optimization for EVA-MVC

As mentioned earlier, EVA can be followed by any MVC method. In our framework EVA-MVC, the widely recognized MSL method OMSC [8] is chosen as the default. In this subsection, OMSC is employed as an illustrative example to showcase the Optimization and Convergence of the proposed framework.

*6.4.1 Overall Framework.* The overall objective function of of integrating EVA with OMSC can be formulated as follows:

$$\min_{\alpha,\beta,U_c^{(v)},Y_c,H_c,A,S,L,F} \sum_{c=1}^{C} \sum_{v=1}^{V^{(c)}} \frac{\alpha_v^2}{2} \|X^{(v)} - U_c^{(v)}Y_c\|_F^2$$

$$+ \frac{\beta_c^2}{2} \|Y_c - H_cAS\|_F^2 + \lambda\|S - LF\|_F^2,$$

$$s.t. \ \alpha^T\mathbf{1} = 1, \alpha \geq 0, Y_c(Y_c)^T = I_{d_c}, (H_c)^T(H_c) = I_l, A^TA = I_r, S^T\mathbf{1} = 1$$

$$L^TL = I_k, F_{ij} \in \{0,1\}, \sum_{i=1}^{k} F_{ij} = 1, \forall j = 1, 2, ..., n, (\beta^2)^T\mathbf{1} = 1, \beta \geq 0.$$

*6.4.2 Optimization of EVA-MVC.* **Initialization.** The variables are initialized as follows: $\alpha = \frac{1}{V^{(c)}}$, $\beta = \frac{1}{\sqrt{C}}$ and $U_c^{(v)} = I \in \mathbb{R}^{m^{(v)} \times k}, Y_c = I \in \mathbb{R}^{d_c \times n}, H_c = I \in \mathbb{R}^{d_c \times l}, A = I \in \mathbb{R}^{l \times r}, S = I \in \mathbb{R}^{r \times n}, L = I \in \mathbb{R}^{r \times k}, F = I \in \mathbb{R}^{k \times n}$, where $k$ is the number of clusters and $I$ denotes the identity matrix. The initialization aims to satisfy the orthogonal constraint.

**Update $U_c^{(v)}$.** To update $U_c^{(v)}$, we fix the other variables and reformulate the overall optimization problem, Eq. 6.4.1, as follows:

$$\min \|X^{(v)} - U_c^{(v)}Y_c\|_F^2. \quad (19)$$

To transform Eq. 19, we use the expanded trace of the Frobenius norm and obtain the following model:

$$\min \ Tr(-2(X^{(v)})^T U_c^{(v)} Y_c + (Y_c)^T (U_c^{(v)})^T U_c^{(v)} Y_c). \quad (20)$$

By setting the derivative of Eq. 20 to zero, $U_c^{(v)}$ is updated as:

$$U_c^{(v)} = X^{(v)}(Y_c)^T. \quad (21)$$

**Update $Y_c$.** Similar to $U_c^{(v)}$, the optimization of $Y_c$ is equivalent to maximizing the following form:

$$\max \ Tr(Y_cB), \quad (22)$$

where $B = \sum_{v=1}^{V^{(g)}} \alpha_v^2 (X^{(v)})^T U_c^{(v)} + \beta_c^2 S^T A^T (H_c)^T$. According to [50], we employ Singular Value Decomposition (SVD) to update $Y_c$. The formula is as follows:

$$Y_c = U_B V_B, \quad (23)$$

where $B = U_B \Sigma_B V_B$.

**Update $H_c$.** Similarly to updating $Y_c^{(v)}$, it is equivalent to solving $H_c$ by the following form:

$$\max \ Tr(H_cW), \quad (24)$$

where $W = Y_c S^T A^T$. Therefore, the optimal solution of variable $H_c = U_W V_W$, where $W = U_W \Sigma_W V_W$.

**Update $A$.** Similar to updating $H_c$, the formula for updating $A$ is as follows:

$$A = U_R V_R, \quad (25)$$

where $R = \sum_{g=1}^{G} \beta_g^2 (H_c)^T P_g S^T$ and $R$ can be decomposed to $U_R \Sigma_R V_R$ by SVD.

**Update $S$.** By fixing the other variables, we can obtain the overall following optimization problem for updating $S$ as follows:

$$\min \sum_{c=1}^{C} \frac{\beta_c^2}{2} \|Y_c - H_cAS\|_F^2 + \lambda\|S - LF\|_F^2. \quad (26)$$

$$s.t. \ S \geq 0, S\mathbf{1} = 1$$

In order to address the optimization problem of $S$, we can reformulate it as a Quadratic Programming (QP) problem [30] as follows:

$$\min \frac{1}{2} S_{:,j}^T Q S_{:,j} + h^T S_{:,j}, \quad (27)$$

where $Q = 2(\sum_{c=1}^{C} \beta_c^2 + \lambda)I$ and $h^T = -2\sum_{c=1}^{G} (Y_c)_{:,j}^T H_c A - 2\lambda F_{:,j}^T G^T$. Therefore each column in matrix $S$ is subsequently regarded as an autonomous QP problem for resolution.

**Update $L$.** Similar to resolution of $H_c$ and $A$, $L$ can be updated by using the following formula:

$$\max \ Tr(L^TD), s.t. \ L^TL = I_k, \quad (28)$$

where $D = SF^T$. Therefore the solution of $L$ is $U_D V_D$, where $D = U_D \Sigma_D V_D$.

**Update $F$.** By fixing the other variables, we can obtain the overall following optimization problem for updating $F$ as follows:

$$\min \lambda\|S - LF\|_F^2, \quad (29)$$

$$s.t. \ F_{ij} = \{0,1\}, \sum_{i=1}^{k} F_{ij} = 1, \forall j = 1, 2, \ldots, n.$$

Notice that in each column of $F$ there is only one 1 and other elements are zeros. Therefore, we can solve Eq. 29 column by column. When solving the $i$-th column, we replace the $i$-th column

by $[1, 0, \ldots, 0]^T$, $[0, 1, 0, \ldots, 0]^T$, $\ldots$, $[0, \ldots, 0, 1]^T$ respectively, and select the one has the lowest objective function value as the solution of the $i$-th row.

**Update $\alpha_v$ and $\beta_c$.** By fixing other variables, the overall objective formulation for updating $\alpha_v$ can be rewritten as:

$$\min \sum_{v=1}^{V^{(c)}} \alpha_v^2 s_v^2, s.t. \ \alpha^T \mathbf{1} = 1, \alpha \geq 0 \quad (30)$$

where $s_v = \|X^{(v)} - U_c^{(v)} Y_c\|_F$. According to Cauchy-Buniakowsky-Schwarz inequality [1], we can update $\alpha_v$ and $\beta_c$ as follows:

$$\alpha_v = \frac{\frac{1}{s_v}}{\sum_v \frac{1}{s_v}}, \beta_c = \frac{\frac{1}{l_c}}{\sum_c \frac{1}{l_c}}, \quad (31)$$

where $l_c = \frac{1}{2}\|Y_c - H_c AS\|_F$.

## 6.5 Framework Convergence Analysis

*6.5.1 Convergence Analysis of EVA-MVC.* In this section, we present the convergence proof for Algorithm 1 (EVA-MVC) in the main text.

THEOREM 6.2. *The proposed EVA-MVC algorithm is proven to be converged.*

PROOF. We begin by defining the objective function of the proposed Algorithm 1 in the main text as follows:

$$\mathcal{J}(\alpha, \beta, U_c^{(v)}, Y_c, H_c, A, S, L, F) =$$

$$min_{\alpha,\beta,U_c^{(v)},Y_c,H_c,A,S,L,F} \sum_p \sum_v \frac{\alpha_v^2}{2} \|X^{(v)} - U_c^{(v)} Y_c\|_F^2$$

$$+ \frac{\beta_c^2}{2}\|Y_c - H_c AS\|_F^2 + \lambda\|S - LF\|_F^2,$$

$$s.t. \ \alpha^T \mathbf{1} = 1, \alpha \geq 0, Y_c(Y_c)^T = I_{d_c}, (H_c)^T(H_c) = I_l, A^T A = I_r, S^T \mathbf{1} = 1,$$

$$L^T L = I_k, F_{ij} \in \{0, 1\}, \sum_{i=1}^k F_{ij} = 1, \forall j = 1, 2, ..., n, \sum_{g=1}^G \beta_c^2 = 1, \beta \geq 0,$$

As observed from the equation above, the entire function is not jointly convex when all variables are considered simultaneously. Instead, we propose an alternate optimization algorithm to optimize each variable while keeping the others fixed. Let we define $\alpha^{(t)}, \beta^{(t)}, (U_c^{(v)})^{(t)}, Y_c^{(t)}, H_c^{(t)}, A^{(t)}, S^{(t)}, L^{(t)}, F^{(t)}$ be the solution at the $t$-iteration.

**(i) Optimizing $U_c^{(v)}$ with fixed** $\alpha, \beta, Y_c, H_c, A, S, L$, **and** $F$. Given $\alpha^{(t)}, \beta^{(t)}, Y_c^{(t)}, H_c^{(t)}, A^{(t)}, S^{(t)}, L^{(t)}, F^{(t)}$, the optimization in Equation 14 (in the main text) respect to $U_c^{(v)}$ can be analytically obtained. The detailed derivation can be found in the Section 3.3 in the main text. Suppose the obtained optimal solution is $(U_c^{(v)})^{(t+1)}$. We have:

$$\mathcal{J}(\alpha^{(t)}, \beta^{(t)}, (U_c^{(v)})^{(t)}, Y_c^{(t)}, H_c^{(t)}, A^{(t)}, S^{(t)}, L^{(t)}, F^{(t)})$$

$$\geq \mathcal{J}(\alpha^{(t)}, \beta^{(t)}, (U_c^{(v)})^{(t+1)}, Y_c^{(t)}, H_c^{(t)}, A^{(t)}, S^{(t)}, L^{(t)}, F^{(t)}). \quad (32)$$

**(ii) Optimizing $Y_c$ with fixed** $\alpha, \beta, U_c^{(v)}, H_c, A, S, L$, **and** $F$. Given $\alpha^{(t)}, \beta^{(t)}, (U_c^{(v)})^{(t)}, H_c^{(t)}, A^{(t)}, S^{(t)}, G^{(t)}$, and $F^{(t)}$, the optimization in Eq. 6.5.1 respect to $Y_c^{(t)}$ can be analytically obtained. Suppose the obtained optimal solution is $Y_c^{(t+1)}$. We have:

$$\mathcal{J}(\alpha^{(t)}, \beta^{(t)}, (U_c^{(v)})^{(t)}, Y_c^{(t)}, H_c^{(t)}, A^{(t)}, S^{(t)}, L^{(t)}, F^{(t)})$$

$$\geq \mathcal{J}(\alpha^{(t)}, \beta^{(t)}, (U_c^{(v)})^{(t)}, Y_c^{(t+1)}, H_c^{(t)}, A^{(t)}, S^{(t)}, L^{(t)}, F^{(t)}). \quad (33)$$

The process of optimizing other variables is the same as **(i)** and **(ii)**. Together with these equations, we have:

$$\mathcal{J}(\alpha^{(t)}, \beta^{(t)}, (U_c^{(v)})^{(t)}, Y_c^{(t)}, H_c^{(t)}, A^{(t)}, S^{(t)}, L^{(t)}, F^{(t)}) \geq$$

$$\mathcal{J}(\alpha^{(t+1)}, \beta^{(t+1)}, (U_c^{(v)})^{(t+1)}, Y_c^{(t+1)}, H_c^{(t+1)}, A^{(t+1)}, S^{(t+1)}, L^{(t+1)}, F^{(t+1)}), \quad (34)$$

which indicates that the objective function of our algorithm monotonically decreases with the increase of iterations. Also, the objective function is lower bounded by zero. As a result, the proposed algorithm can be verified to converge to a local minimum. □

According to Theorem 6.2 and [2], our proposed algorithm is theoretically guaranteed to converge to a local minimum. Furthermore, the conducted experiments on benchmarks demonstrate the convergence of EVA-MVC.

## 6.6 Comprehensive Experiments

*6.6.1 Generalization Evaluation.* We evaluate the generalization capability of EVA-MVC by integrating EVA with representative Multi-view Clustering (MVC) methods across various categories (as detailed in Table 1). The complete results are presented in Table 5.

**Table 5: Generalization Evaluation on EVA-MVC. (Metric: ACC)**

| Datasets | 3Sources | Caltech101-7 | YTF10 | YTF20 | YTF50 |
|---|---|---|---|---|---|
| FPMVS | 0.4201 | 0.6872 | 0.7326 | 0.6948 | 0.6851 |
| EVA+FPMVS | **0.6508** | **0.7001** | **0.7588** | **0.7300** | **0.6719** |
| MSGL | 0.3353 | **0.6282** | **0.7549** | 0.5978 | 0.4675 |
| EVA+MSGL | **0.3491** | 0.5421 | 0.7249 | **0.6768** | **0.6675** |
| SMVSC | 0.3786 | **0.7014** | 0.7406 | 0.6517 | 0.6393 |
| EVA+SMVSC | **0.7100** | 0.7001 | **0.7749** | **0.7438** | **0.6906** |
| OMSC | 0.3372 | 0.6614 | 0.7820 | 0.7446 | 0.7152 |
| EVA+OMSC | **0.7692** | **0.8853** | **0.7854** | **0.7776** | **0.7918** |
| FMVACC | 0.4923 | 0.4105 | 0.7141 | 0.6883 | 0.6704 |
| EVA+FMVACC | **0.8047** | **0.4769** | **0.7986** | **0.7269** | **0.6844** |
| AWMVC | 0.6450 | 0.3693 | 0.7140 | 0.6395 | 0.6671 |
| EVA+AWMVC | **0.7337** | **0.5413** | **0.7551** | **0.7762** | **0.7166** |
| FastMICE | 0.5041 | 0.5362 | **0.7703** | 0.6797 | 0.6921 |
| EVA+FastMICE | **0.7443** | **0.5785** | 0.7511 | 0.6768 | **0.6999** |
| EMVGC-LG | 0.5321 | 0.3708 | 0.7755 | 0.7279 | 0.6515 |
| EVA+EMVGC-LG | **0.5628** | **0.4113** | **0.7836** | **0.7326** | **0.6725** |
| CMVC | 0.7396 | 0.5039 | 0.8120 | 0.7588 | 0.7138 |
| EVA+CMVC | **0.7633** | **0.5339** | **0.9052** | **0.7761** | **0.7436** |

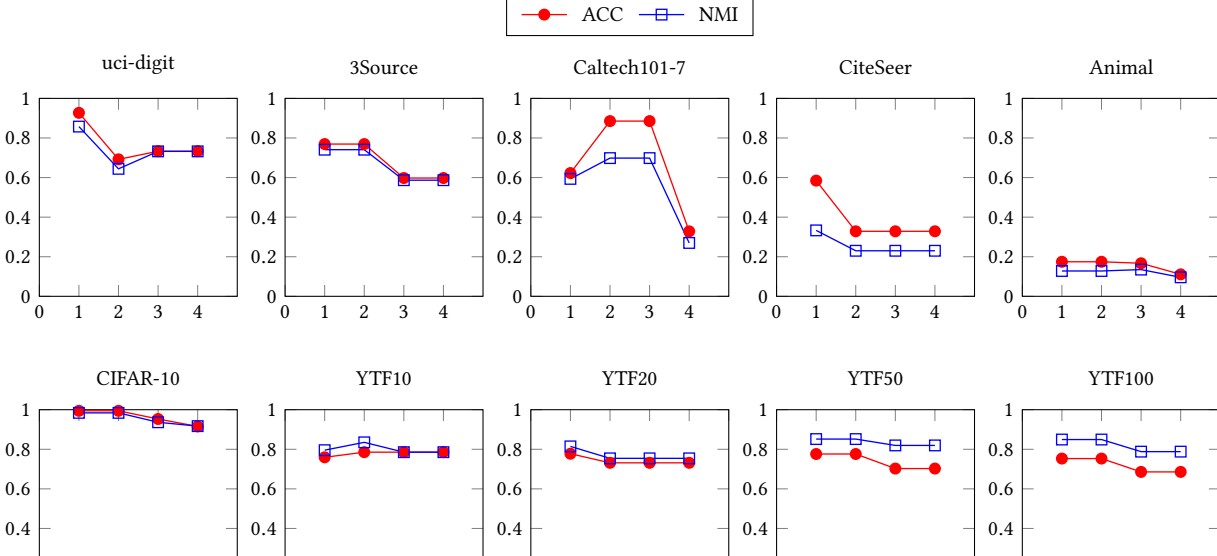

Figure 10: Parameter analysis on $\tau$ (*x*-axis)

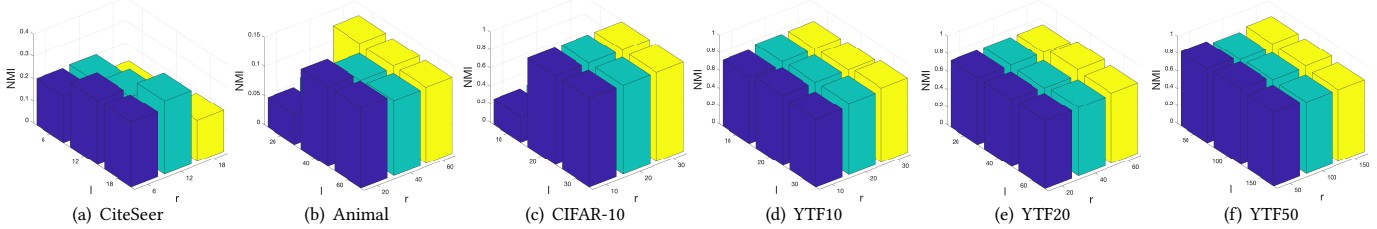

(a) CiteSeer  (b) Animal  (c) CIFAR-10  (d) YTF10  (e) YTF20  (f) YTF50

Figure 11: Parameter *l* (*x*-axis) and *r* (*y*-axis) analysis

*6.6.2  Parameter Analysis.* The comprehensive results of the Parameter Analysis are depicted in Figure 10 and Figure 11.

*6.6.3  Analysis on View Supplementary Graph (VSG) and Decision Map .* The VSG plots of multi-view datasets are presented in Figure 12, while the corresponding decision maps are displayed in Figure 13. In the VSG plots, datasets like Caltech101-7, YTF10, YTF20, YTF50, and YTF100 reveal the issue of "Redundant Supplementarity", showcasing significant overlaps among the views.

Concerning the decision maps, the decision boundaries are uniformly set at $\tau = 1.5 \pm 0.3$ for all datasets.

*6.6.4  Convergence Analysis.* The assessment of framework convergence is depicted in Figure 14.

*6.6.5  Performance Comparison.* A detailed performance comparison is provided in Table 6.

In summary, the conclusions derived from the comprehensive experiments above align with those drawn in the main text.

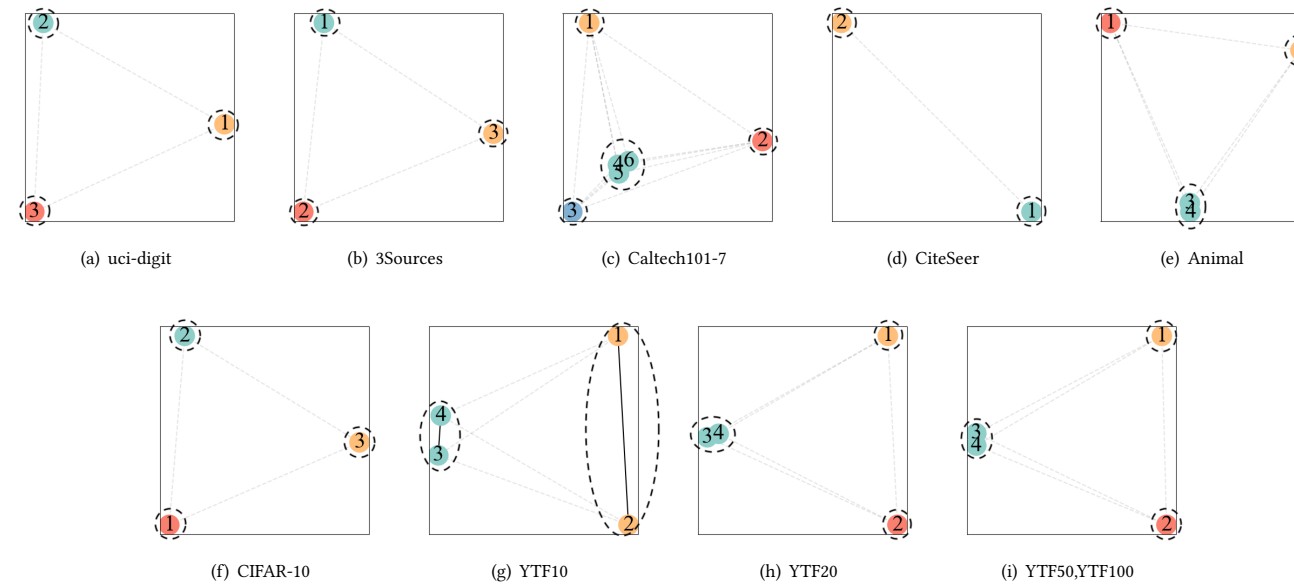

**Figure 12: Illustration of View Supplementarity Graphs (VSGs) in seven multi-view datasets. Each circle represents a specific view, labeled with the corresponding number. The edges between circles indicate the similarity in supplementarity between the views. The black dotted circles represent the View Communities obtained by Decision Maps.**

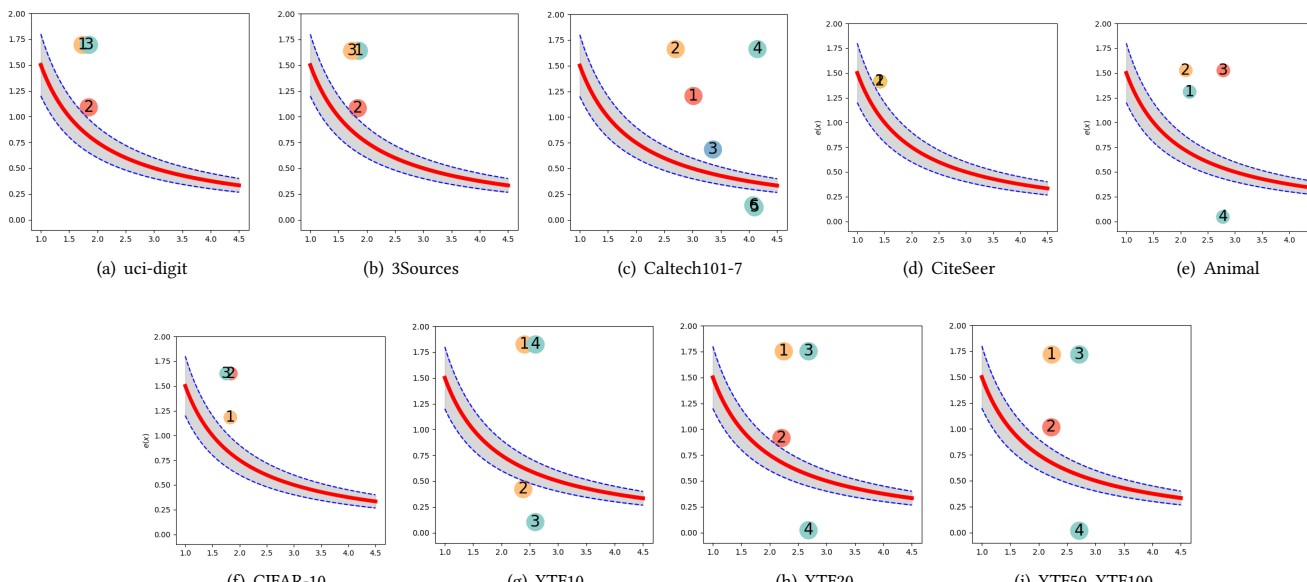

**Figure 13: Decision maps of multi-view datasets, where the $x$-axis represents the density $\rho$ and the $y$-axis indicates the dependent distance $\delta$. The decision boundary is uniformly set as $y = \frac{\tau}{x}$, where $\tau = 1.5 \pm 0.3$.**

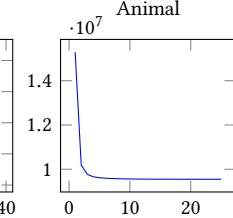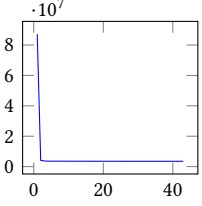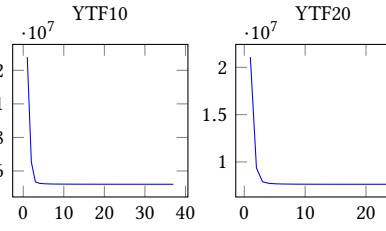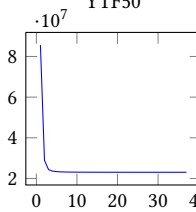

**Figure 14: Convergence analysis (x-axis: the number of Iterations $T$)**

**Table 6: Comparison results.**

| Datasets | Metric | LF-LAM | FPMVS | MSGL | SMVSC | OMSC | FMVACC | AWMVC | FastMICE | EMVGC-LG | CMVC | EVA-MVC |
|---|---|---|---|---|---|---|---|---|---|---|---|---|
| uci-digit | ACC | $0.9005_{\pm0.014}$ | $0.8265_{\pm0.000}$ | $0.7380_{\pm0.029}$ | $0.8055_{\pm0.000}$ | $0.7325_{\pm0.000}$ | $0.7716_{\pm0.062}$ | $0.7749_{\pm0.015}$ | $0.8070_{\pm0.039}$ | $0.8889_{\pm0.025}$ | $0.9205_{\pm0.000}$ | $\mathbf{0.9270_{\pm0.000}}$ |
| | NMI | $0.8186_{\pm0.017}$ | $0.8124_{\pm0.000}$ | $0.7417_{\pm0.016}$ | $0.7557_{\pm0.000}$ | $0.7490_{\pm0.000}$ | $0.7339_{\pm0.030}$ | $0.7427_{\pm0.009}$ | $0.8185_{\pm0.028}$ | $0.8380_{\pm0.017}$ | $0.8576_{\pm0.000}$ | $\mathbf{0.8576_{\pm0.000}}$ |
| | Purity | $0.9005_{\pm0.014}$ | $0.8270_{\pm0.000}$ | $0.8120_{\pm0.027}$ | $0.8055_{\pm0.000}$ | $0.7395_{\pm0.000}$ | $0.7933_{\pm0.047}$ | $0.7935_{\pm0.014}$ | $0.8603_{\pm0.039}$ | $0.8932_{\pm0.023}$ | $0.9205_{\pm0.000}$ | $\mathbf{0.9270_{\pm0.000}}$ |
| | Fscore | $0.8160_{\pm0.022}$ | $0.7733_{\pm0.000}$ | $0.6606_{\pm0.022}$ | $0.7003_{\pm0.000}$ | $0.6848_{\pm0.000}$ | $0.6923_{\pm0.043}$ | $0.6987_{\pm0.013}$ | $0.8054_{\pm0.040}$ | $0.8182_{\pm0.026}$ | $0.8517_{\pm0.000}$ | $\mathbf{0.8621_{\pm0.000}}$ |
| 3Source | ACC | $0.5207_{\pm0.009}$ | $0.4201_{\pm0.000}$ | $0.3353_{\pm0.012}$ | $0.3786_{\pm0.000}$ | $0.3372_{\pm0.000}$ | $0.4923_{\pm0.044}$ | $0.6450_{\pm0.010}$ | $0.5041_{\pm0.054}$ | $0.5321_{\pm0.050}$ | $0.7396_{\pm0.000}$ | $\mathbf{0.7692_{\pm0.000}}$ |
| | NMI | $0.5110_{\pm0.010}$ | $0.1578_{\pm0.000}$ | $0.0630_{\pm0.013}$ | $0.1117_{\pm0.000}$ | $0.1271_{\pm0.000}$ | $0.3792_{\pm0.051}$ | $0.5419_{\pm0.007}$ | $0.4135_{\pm0.053}$ | $0.4778_{\pm0.048}$ | $0.6566_{\pm0.000}$ | $\mathbf{0.7408_{\pm0.000}}$ |
| | Purity | $0.7337_{\pm0.015}$ | $0.5325_{\pm0.000}$ | $0.6331_{\pm0.135}$ | $0.4378_{\pm0.000}$ | $0.4378_{\pm0.000}$ | $0.6047_{\pm0.040}$ | $0.7343_{\pm0.006}$ | $0.6177_{\pm0.019}$ | $0.6608_{\pm0.038}$ | $0.7870_{\pm0.000}$ | $\mathbf{0.8579_{\pm0.000}}$ |
| | Fscore | $0.4822_{\pm0.015}$ | $0.3391_{\pm0.000}$ | $0.3713_{\pm0.026}$ | $0.2914_{\pm0.000}$ | $0.2530_{\pm0.000}$ | $0.4100_{\pm0.043}$ | $0.5633_{\pm0.009}$ | $0.4022_{\pm0.054}$ | $0.4590_{\pm0.047}$ | $0.7079_{\pm0.000}$ | $\mathbf{0.7531_{\pm0.000}}$ |
| Caltech101-7 | ACC | $0.4322_{\pm0.027}$ | $0.6872_{\pm0.000}$ | $0.6282_{\pm0.088}$ | $0.7014_{\pm0.000}$ | $0.6614_{\pm0.000}$ | $0.4105_{\pm0.017}$ | $0.3693_{\pm0.008}$ | $0.5362_{\pm0.011}$ | $0.3708_{\pm0.017}$ | $0.5039_{\pm0.000}$ | $\mathbf{0.8853_{\pm0.000}}$ |
| | NMI | $0.5091_{\pm0.030}$ | $0.5055_{\pm0.000}$ | $0.4448_{\pm0.114}$ | $0.5633_{\pm0.000}$ | $0.5567_{\pm0.000}$ | $0.3939_{\pm0.017}$ | $0.4957_{\pm0.009}$ | $0.5778_{\pm0.013}$ | $0.4870_{\pm0.015}$ | $0.5647_{\pm0.000}$ | $\mathbf{0.6983_{\pm0.000}}$ |
| | Purity | $0.8541_{\pm0.020}$ | $0.8086_{\pm0.000}$ | $0.7069_{\pm0.067}$ | $0.8656_{\pm0.000}$ | $0.8588_{\pm0.000}$ | $0.7734_{\pm0.022}$ | $0.8348_{\pm0.005}$ | $0.6160_{\pm0.018}$ | $0.8226_{\pm0.007}$ | $0.8559_{\pm0.000}$ | $\mathbf{0.9084_{\pm0.000}}$ |
| | Fscore | $0.4529_{\pm0.008}$ | $0.6728_{\pm0.000}$ | $0.5956_{\pm0.063}$ | $0.6809_{\pm0.000}$ | $0.6503_{\pm0.000}$ | $0.4114_{\pm0.018}$ | $0.4364_{\pm0.007}$ | $0.5714_{\pm0.014}$ | $0.4135_{\pm0.012}$ | $0.5485_{\pm0.000}$ | $\mathbf{0.8615_{\pm0.000}}$ |
| CiteSeer | ACC | $0.3897_{\pm0.022}$ | $0.3867_{\pm0.000}$ | $0.2137_{\pm0.005}$ | $0.3734_{\pm0.000}$ | $0.3867_{\pm0.000}$ | $0.4480_{\pm0.074}$ | $0.4300_{\pm0.055}$ | $0.4355_{\pm0.023}$ | $0.3969_{\pm0.091}$ | $0.5284_{\pm0.000}$ | $\mathbf{0.5845_{\pm0.000}}$ |
| | NMI | $0.1604_{\pm0.028}$ | $0.1439_{\pm0.000}$ | $0.0109_{\pm0.004}$ | $0.1486_{\pm0.000}$ | $0.1472_{\pm0.000}$ | $0.2280_{\pm0.060}$ | $0.2332_{\pm0.014}$ | $0.2195_{\pm0.025}$ | $0.2272_{\pm0.074}$ | $0.2607_{\pm0.000}$ | $\mathbf{0.3331_{\pm0.000}}$ |
| | Purity | $0.4142_{\pm0.021}$ | $0.4060_{\pm0.000}$ | $0.2171_{\pm0.005}$ | $0.4018_{\pm0.000}$ | $0.4344_{\pm0.000}$ | $0.4849_{\pm0.077}$ | $0.4943_{\pm0.037}$ | $0.4905_{\pm0.023}$ | $0.4172_{\pm0.086}$ | $0.5501_{\pm0.000}$ | $\mathbf{0.6237_{\pm0.000}}$ |
| | Fscore | $0.2869_{\pm0.017}$ | $0.2908_{\pm0.000}$ | $0.2965_{\pm0.012}$ | $0.2853_{\pm0.000}$ | $0.2916_{\pm0.000}$ | $0.3340_{\pm0.051}$ | $0.3403_{\pm0.024}$ | $0.3237_{\pm0.017}$ | $0.3240_{\pm0.029}$ | $0.3769_{\pm0.000}$ | $\mathbf{0.4451_{\pm0.000}}$ |
| Animal | ACC | N/A | $\mathbf{0.2026_{\pm0.000}}$ | $0.1350_{\pm0.005}$ | $0.1740_{\pm0.000}$ | $0.1804_{\pm0.000}$ | $0.1334_{\pm0.002}$ | $0.1494_{\pm0.004}$ | $0.1641_{\pm0.003}$ | $0.1765_{\pm0.008}$ | $0.1727_{\pm0.000}$ | $0.1883_{\pm0.000}$ |
| | NMI | N/A | $\mathbf{0.1596_{\pm0.000}}$ | $0.0935_{\pm0.004}$ | $0.1444_{\pm0.000}$ | $0.1434_{\pm0.000}$ | $0.0896_{\pm0.001}$ | $0.1246_{\pm0.003}$ | $0.1299_{\pm0.004}$ | $0.1413_{\pm0.004}$ | $0.1541_{\pm0.000}$ | $0.1503_{\pm0.000}$ |
| | Purity | N/A | $0.2124_{\pm0.000}$ | $0.1742_{\pm0.006}$ | $0.2045_{\pm0.000}$ | $0.2050_{\pm0.000}$ | $0.1672_{\pm0.002}$ | $0.1869_{\pm0.005}$ | $0.1858_{\pm0.005}$ | $0.2068_{\pm0.009}$ | $0.2190_{\pm0.000}$ | $\mathbf{0.2207_{\pm0.000}}$ |
| | Fscore | N/A | $\mathbf{0.1466_{\pm0.000}}$ | $0.0978_{\pm0.003}$ | $0.1045_{\pm0.000}$ | $0.1314_{\pm0.000}$ | $0.0825_{\pm0.001}$ | $0.0975_{\pm0.001}$ | $0.1020_{\pm0.001}$ | $0.1139_{\pm0.003}$ | $0.1099_{\pm0.000}$ | $0.1117_{\pm0.000}$ |
| CIFAR-10 | ACC | N/A | $0.9898_{\pm0.000}$ | $0.9314_{\pm0.028}$ | $0.9882_{\pm0.000}$ | $0.9885_{\pm0.000}$ | $0.9535_{\pm0.049}$ | $0.9282_{\pm0.090}$ | $0.9500_{\pm0.056}$ | $0.9154_{\pm0.045}$ | $0.9931_{\pm0.000}$ | $\mathbf{0.9944_{\pm0.000}}$ |
| | NMI | N/A | $0.9729_{\pm0.000}$ | $0.8843_{\pm0.034}$ | $0.9690_{\pm0.000}$ | $0.9697_{\pm0.000}$ | $0.9365_{\pm0.017}$ | $0.9112_{\pm0.032}$ | $0.9625_{\pm0.018}$ | $0.9178_{\pm0.018}$ | $0.9811_{\pm0.000}$ | $\mathbf{0.9841_{\pm0.000}}$ |
| | Purity | N/A | $0.9898_{\pm0.000}$ | $0.9314_{\pm0.027}$ | $0.9882_{\pm0.000}$ | $0.9885_{\pm0.000}$ | $0.9541_{\pm0.048}$ | $0.9394_{\pm0.064}$ | $0.9899_{\pm0.001}$ | $0.9214_{\pm0.037}$ | $0.9931_{\pm0.000}$ | $\mathbf{0.9944_{\pm0.000}}$ |
| | Fscore | N/A | $0.9800_{\pm0.000}$ | $0.8763_{\pm0.023}$ | $0.9767_{\pm0.000}$ | $0.9773_{\pm0.000}$ | $0.9360_{\pm0.043}$ | $0.9094_{\pm0.067}$ | $0.9463_{\pm0.050}$ | $0.8995_{\pm0.041}$ | $0.9864_{\pm0.000}$ | $\mathbf{0.9890_{\pm0.000}}$ |
| YTF10 | ACC | N/A | $0.7326_{\pm0.000}$ | $0.7549_{\pm0.056}$ | $0.7406_{\pm0.000}$ | $0.7820_{\pm0.000}$ | $0.7141_{\pm0.054}$ | $0.7104_{\pm0.017}$ | $0.7703_{\pm0.050}$ | $0.7755_{\pm0.024}$ | $0.8120_{\pm0.000}$ | $\mathbf{0.7854_{\pm0.000}}$ |
| | NMI | N/A | $0.7740_{\pm0.000}$ | $0.7923_{\pm0.042}$ | $0.7794_{\pm0.000}$ | $0.8275_{\pm0.000}$ | $0.7541_{\pm0.019}$ | $0.7830_{\pm0.009}$ | $0.8068_{\pm0.020}$ | $0.8067_{\pm0.016}$ | $0.8349_{\pm0.000}$ | $\mathbf{0.8351_{\pm0.000}}$ |
| | Purity | N/A | $0.7621_{\pm0.000}$ | $\mathbf{0.8546_{\pm0.025}}$ | $0.7619_{\pm0.000}$ | $0.7810_{\pm0.000}$ | $0.7529_{\pm0.034}$ | $0.7935_{\pm0.010}$ | $0.8518_{\pm0.026}$ | $0.7967_{\pm0.018}$ | $0.8469_{\pm0.000}$ | $0.8327_{\pm0.000}$ |
| | Fscore | N/A | $0.6960_{\pm0.000}$ | $0.6489_{\pm0.084}$ | $0.6835_{\pm0.000}$ | $0.7456_{\pm0.000}$ | $0.6761_{\pm0.035}$ | $0.6875_{\pm0.014}$ | $0.7475_{\pm0.034}$ | $0.7366_{\pm0.031}$ | $0.7702_{\pm0.000}$ | $\mathbf{0.7788_{\pm0.000}}$ |
| YTF20 | ACC | N/A | $0.6948_{\pm0.000}$ | $0.5978_{\pm0.039}$ | $0.6517_{\pm0.000}$ | $0.7446_{\pm0.000}$ | $0.6883_{\pm0.024}$ | $0.6395_{\pm0.013}$ | $0.6797_{\pm0.032}$ | $0.7279_{\pm0.028}$ | $0.7588_{\pm0.000}$ | $\mathbf{0.7776_{\pm0.000}}$ |
| | NMI | N/A | $0.7740_{\pm0.000}$ | $0.7166_{\pm0.026}$ | $0.7586_{\pm0.000}$ | $\mathbf{0.8170_{\pm0.000}}$ | $0.7712_{\pm0.014}$ | $0.7669_{\pm0.006}$ | $0.8007_{\pm0.014}$ | $0.7901_{\pm0.005}$ | $0.7861_{\pm0.000}$ | $0.8146_{\pm0.000}$ |
| | Purity | N/A | $0.7259_{\pm0.000}$ | $0.7516_{\pm0.035}$ | $0.7045_{\pm0.000}$ | $0.7731_{\pm0.000}$ | $0.7249_{\pm0.028}$ | $0.7108_{\pm0.010}$ | $0.7704_{\pm0.017}$ | $0.7427_{\pm0.008}$ | $0.7850_{\pm0.000}$ | $\mathbf{0.8066_{\pm0.000}}$ |
| | Fscore | N/A | $0.6261_{\pm0.000}$ | $0.4579_{\pm0.052}$ | $0.6248_{\pm0.000}$ | $0.6835_{\pm0.000}$ | $0.6355_{\pm0.025}$ | $0.5976_{\pm0.016}$ | $0.5980_{\pm0.061}$ | $0.6438_{\pm0.016}$ | $0.6595_{\pm0.000}$ | $\mathbf{0.7147_{\pm0.000}}$ |
| YTF50 | ACC | N/A | $0.6851_{\pm0.000}$ | $0.4675_{\pm0.042}$ | $0.6393_{\pm0.000}$ | $0.7152_{\pm0.000}$ | $0.6704_{\pm0.020}$ | $0.6671_{\pm0.007}$ | $0.6921_{\pm0.023}$ | $0.6515_{\pm0.014}$ | $0.7138_{\pm0.000}$ | $\mathbf{0.7918_{\pm0.000}}$ |
| | NMI | N/A | $0.8364_{\pm0.000}$ | $0.6446_{\pm0.049}$ | $0.8019_{\pm0.000}$ | $0.8170_{\pm0.000}$ | $0.8220_{\pm0.007}$ | $0.8308_{\pm0.001}$ | $0.8315_{\pm0.007}$ | $0.7948_{\pm0.003}$ | $0.8375_{\pm0.000}$ | $\mathbf{0.8487_{\pm0.000}}$ |
| | Purity | N/A | $0.7140_{\pm0.000}$ | $0.6516_{\pm0.041}$ | $0.6514_{\pm0.000}$ | $0.7731_{\pm0.000}$ | $0.6967_{\pm0.018}$ | $0.7340_{\pm0.004}$ | $0.7518_{\pm0.016}$ | $0.7307_{\pm0.024}$ | $0.7728_{\pm0.000}$ | $\mathbf{0.8359_{\pm0.000}}$ |
| | Fscore | N/A | $0.6381_{\pm0.000}$ | $0.2849_{\pm0.025}$ | $0.5423_{\pm0.000}$ | $0.6835_{\pm0.000}$ | $0.6087_{\pm0.026}$ | $0.5964_{\pm0.006}$ | $0.6124_{\pm0.023}$ | $0.6142_{\pm0.010}$ | $0.6313_{\pm0.000}$ | $\mathbf{0.7145_{\pm0.000}}$ |
| YTF100 | ACC | N/A | $0.5293_{\pm0.000}$ | $0.4340_{\pm0.035}$ | $0.5906_{\pm0.000}$ | $0.6651_{\pm0.000}$ | $0.6344_{\pm0.002}$ | $0.6283_{\pm0.010}$ | $0.6683_{\pm0.016}$ | $0.6195_{\pm0.014}$ | $0.6652_{\pm0.000}$ | $\mathbf{0.7531_{\pm0.000}}$ |
| | NMI | N/A | $0.7532_{\pm0.000}$ | $0.6342_{\pm0.046}$ | $0.7991_{\pm0.000}$ | $0.8337_{\pm0.000}$ | $0.8190_{\pm0.004}$ | $0.8304_{\pm0.002}$ | $0.8309_{\pm0.069}$ | $0.8247_{\pm0.005}$ | $0.8318_{\pm0.000}$ | $\mathbf{0.8492_{\pm0.000}}$ |
| | Purity | N/A | $0.5446_{\pm0.000}$ | $0.6100_{\pm0.038}$ | $0.6103_{\pm0.000}$ | $0.7141_{\pm0.000}$ | $0.6659_{\pm0.020}$ | $0.7212_{\pm0.007}$ | $0.7359_{\pm0.014}$ | $0.7163_{\pm0.008}$ | $0.7375_{\pm0.000}$ | $\mathbf{0.8000_{\pm0.000}}$ |
| | Fscore | N/A | $0.3541_{\pm0.000}$ | $0.1562_{\pm0.033}$ | $0.5035_{\pm0.000}$ | $0.5846_{\pm0.000}$ | $0.5765_{\pm0.026}$ | $0.5297_{\pm0.007}$ | $0.6000_{\pm0.022}$ | $0.5147_{\pm0.006}$ | $0.5850_{\pm0.000}$ | $\mathbf{0.7043_{\pm0.000}}$ |

