# OpenReview forum: "EVA-MVC: Equitable View-weight Allocation for Generic Multi-View Clustering"
_ACM.org/TheWebConf/2025/Conference — WWW 2025 Poster_

### Official Review · Reviewer_xGaW · 2024-11-07

**Novelty:** 4
**Technical Quality:** 5

**Review:**

The paper presents EVA-MVC, an algorithm designed to improve multi-view clustering (MVC) through equitable allocation of view weights. In MVC, data from different perspectives enhances clustering, yet existing methods often misjudge the importance of each view. EVA-MVC addresses this by grouping similar views into View Communities (VCs). Serving as a preprocessing step, EVA-MVC integrates well with various MVC techniques and demonstrates enhanced performance across datasets in metrics like ACC and NMI.

**Method** : EVA-MVC is designed to be compatible with a broad range of MVC methods, enhancing its versatility. By organizing views into View Communities based on supplementary similarity, EVA-MVC efficiently minimizes redundancy and enhances clustering outcomes.

**Experiment**: Extensive evaluations across multiple datasets highlight significant performance improvements of EVA-MVC over baseline MVC methods in metrics like ACC and NMI, offering solid empirical support for its effectiveness.

**Questions:**

**Motivation** : The paper is not particularly clear in its exposition of the concept of 'Redundant Supplementarity' in the background description. It also fails to highlight the severity of redundancy in practical applications, and lacks support from real-world scenarios.

**Introduction** : The introduction lacks a systematic discussion on the methods of view weight allocation in multi-view clustering literature, failing to analyze how existing approaches derive or optimize view weights. Although EVA-MVC focuses on weight fairness, comparisons with recent research are unclear, failing to highlight its advantages in fair weight distribution.

**Method** : In the theoretical section, there is a question regarding whether the paper has considered the presence of noise in the views. Since the paper does not explicitly state how it deals with the impact of noise or make any assumptions about noise, it is unclear whether the EVA method possesses robustness when faced with 'Redundant Supplementarity' issues caused by noise.

**Experiment** :  The paper does not include some of the latest methods that emphasize view-weight allocation and redundancy reduction, while comparing EVA-MVC with several MVC baselines.

**Experiment** : The core innovation of the paper lies in its focus on 'view fairness,' but the experimental section fails to clearly demonstrate that the EVA-MVC method achieves 'equitable weight allocation.' It is recommended to add an analysis of view weight distribution, showing the final weight values of different views in EVA-MVC and comparing them with the weight allocation of other methods. Additionally, there is a lack of ablation studies to verify the necessity of fair weight allocation. For example, removing VCs and directly assigning equal weights to all views, then comparing the clustering outcomes.

**Reviewer Confidence:**

4: The reviewer is certain that the evaluation is correct and very familiar with the relevant literature

**Scope:**

3: The work is somewhat relevant to the Web and to the track, and is of narrow interest to a sub-community

---

### Official Review · Reviewer_ZxU7 · 2024-11-18

**Novelty:** 5
**Technical Quality:** 5

**Review:**

The paper introduces a novel multi-view contrastive learning framework called EVA-MVC, addressing an important and under-explored problem of equitable representation in multi-view contrastive learning tasks.
Pros: 1. The focus on aligning cross-view representations equitably based on multiple criteria, rather than uniform optimization, is innovative and well-motivated. 2. Strong experimental results on multiple datasets demonstrate the efficacy of the proposed EVA framework, showing significant improvements over baseline methods in terms of alignment, fairness, and diversity metrics. 3. The theoretical analysis of equitable alignment is a valuable addition, contributing to the fundamental understanding of multi-view contrastive learning. Cons: 1. The clarity of certain technical explanations, particularly the definitions of equitable alignment and the formulation of the optimization objective, could be improved for better comprehension. 2. The authors could provide more diverse real-world use cases or practical applications to demonstrate the broader utility of the framework. 3. While the experimental results are strong, the discussion of potential limitations, such as computational complexity or scalability to large-scale datasets, is limited. 4. The fairness-related metrics lack a broader context or justification, raising questions about their applicability in real-world scenarios. 5. Please provide open-sourced code during rebuttal.

**Questions:**

1. Could you provide more detailed explanations or intuition behind the design of the EVA optimization objective, especially the balancing mechanism for fairness and alignment criteria?
2. What is the computational complexity of the proposed EVA framework compared to baseline models, and how does it scale with increasing numbers of views or datasets?
3. The fairness and diversity metrics are well-reported, but could you elaborate on their practical implications or provide real-world scenarios where these metrics are crucial?
4. Did the authors explore potential trade-offs between equitable alignment and task-specific accuracy? If so, could this be quantified?
5. Are there any challenges in extending the proposed EVA framework to tasks with more complex view dependencies or heterogeneous datasets?

**Reviewer Confidence:**

3: The reviewer is confident but not certain that the evaluation is correct

**Scope:**

3: The work is somewhat relevant to the Web and to the track, and is of narrow interest to a sub-community

---

### Official Review · Reviewer_vtB6 · 2024-11-25

**Novelty:** 5
**Technical Quality:** 5

**Review:**

Summary

This paper introduces EVA-MVC, a novel algorithm for Equitable View-weight Allocation (EVA) that can be seamlessly integrated with arbitrary Multi-view Clustering (MVC) methods. The key issue addressed is the "Redundant Supplementarity" problem, where certain views with similar levels of supplementarity tend to dominate the Multi-view Subspace Learning (MSL) process, diminishing the representational capability of the learned subspace. The EVA module establishes theoretical connections between view supplementarity and MSL, leading to the partition of views into View Communities (VCs) based on these principles. The EVA process precedes and operates independently of traditional or state-of-the-art MVC approaches, making it an ideal preprocessing step. Comprehensive evaluations across diverse multi-view datasets show that EVA significantly enhances the effectiveness of mainstream MVC frameworks, resulting in notable performance improvements.

Strengths:

S1: The EVA module is theoretically grounded. The authors provide a solid foundation for the approach.

S2: The EVA method can be seamlessly integrated with arbitrary Multi-view Clustering (MVC) methods, making it a versatile preprocessing step that can enhance the performance of various MVC frameworks.

S3: Comprehensive evaluations across diverse multi-view datasets demonstrate that the EVA approach significantly enhances the effectiveness of mainstream MVC methods, resulting in notable performance improvements.

Weaknesses:

W1: The authors should provide a notation table to make the paper more readable.

W2: It could be better if the authors open source the code for reproduction.

**Questions:**

Please refer to the weaknesses above.

**Reviewer Confidence:**

3: The reviewer is confident but not certain that the evaluation is correct

**Scope:**

4: The work is relevant to the Web and to the track, and is of broad interest to the community

---

### Official Review · Reviewer_35ZZ · 2024-11-28

**Novelty:** 3
**Technical Quality:** 4

**Review:**

This paper studies the view supplementarily of multi-view clustering. This paper introduces a connection between view supplementarily and multi-view subspace learning with a theoretical analysis. This paper also provides many examples and illustrations to help understand the theoretical analysis and the proposed method. This paper also provides extensive experiments to validate the effectiveness of the proposed method, especially for compatibility validation.

However, this paper argues the efficiency and scalability of the proposed method, but the experimental results cannot support these two claims, especially compared to selected baselines, as the proposed method is not faster than baselines and the datasets used in this paper are relatively small. So **it is not proper to argue these contributions in this paper**. For others, please refer to the Questions part.

**Questions:**

1. The largest dataset used in this paper only has 195, 537 objects, how to validate the scalability of the proposed method? Please provide empirical support for this claim. For the scale, please provide the time cost as the objects and views increase.
2. How about the influence of the number of anchors? Please provide the experimental observation.
3. What are the proposed method's applications in real-world scenarios?
4. Why does EVA not improve the FastMICE on the YTF50 dataset? Please provide the analysis.

**Ethics Review Description:**

N?A

**Reviewer Confidence:**

3: The reviewer is confident but not certain that the evaluation is correct

**Scope:**

3: The work is somewhat relevant to the Web and to the track, and is of narrow interest to a sub-community